# The NE/AAT/CBG axis regulates adipose tissue glucocorticoid exposure

Luke D. Boyle[1], Allende Miguelez-Crespo[1], Mhairi Paul[1], Elisa Villalobos [1,2], Julia N. C. Toews [3], Lisa Ivatt[1], Boglarka Nagy[1], Marisa Magennis [1], Natalie Z. M. Homer [1], Ruth Andrew [1], Victor Viau[3], Geoffrey L. Hammond[3], Roland H. Stimson [1], Brian R. Walker [1,2] & Mark Nixon [1] ✉

Corticosteroid binding globulin (CBG; *SERPINA6*) binds >85% of circulating glucocorticoids but its influence on their metabolic actions is unproven. Targeted proteolytic cleavage of CBG by neutrophil elastase (NE; *ELANE*) significantly reduces CBG binding affinity, potentially increasing 'free' glucocorticoid levels at sites of inflammation. NE is inhibited by alpha-1-antitrypsin (AAT; *SERPINA1*). Using complementary approaches in mice and humans to manipulate NE or AAT, we show high-fat diet (HFD) increases the NE:AAT ratio specifically in murine visceral adipose tissue, an effect only observed in males. Notably, HFD-fed male mice lacking NE have reduced glucocorticoid levels and action specifically in visceral adipose tissue, with improved glucose tolerance and insulin sensitivity, independent of systemic changes in free glucocorticoids. The protective effect of NE deficiency is lost when the adrenals are removed. Moreover, human asymptomatic heterozygous carriers of deleterious mutations in *SERPINA1* resulting in lower AAT levels have increased adipose tissue glucocorticoid levels and action. However, in contrast to mice, humans present with systemic increases in free circulating glucocorticoid levels, an effect independent of HPA axis activation. These findings show that NE and AAT regulate local tissue glucocorticoid bioavailability in vivo, providing crucial evidence of a mechanism linking inflammation and metabolism.

Glucocorticoids are steroid hormones that critically mediate the body's adaptive response to stress and trauma. Produced by the adrenal gland in response to activation of the hypothalamic-pituitary-adrenal (HPA) axis, they act in multiple tissues via glucocorticoid and mineralocorticoid receptors. A major function of glucocorticoids is to regulate fuel metabolism. In particular, increased adipose tissue glucocorticoid exposure is associated with insulin resistance[1–3], largely attributable to the influence of glucocorticoids on glucose and lipid utilisation. Moreover, hyper-cortisolemia is causally associated with cardiovascular disease[4,5]. Hence there has been significant interest in understanding the

mechanisms that regulate glucocorticoid action within adipose tissue, with a view to therapeutic targeting to reduce cardiometabolic disease.

Evidence in mice has established a role for neutrophil elastase (NE; *ELANE*) in obesity-associated metabolic dysregulation[6–8]. High fat diet-fed mice display rapid adipose tissue neutrophil infiltration and increased adipose NE levels[7], mirroring the higher NE observed in humans with obesity[8]. Notably, mice lacking NE (*Elane*$^{-/-}$) display resistance to diet-induced obesity and improved insulin sensitivity compared to controls[8]. Moreover, treatment of obese mice with alpha-1 anti-trypsin (AAT; *SERPINA1*), an endogenous inhibitor of NE,

[1]University/BHF Centre for Cardiovascular Science, University of Edinburgh, Edinburgh, UK. [2]Translational & Clinical Research Institute, Newcastle University, Newcastle upon Tyne, UK. [3]Life Sciences Centre, University of British Columbia, Vancouver, Canada. ✉e-mail: m.nixon@ed.ac.uk

improves insulin sensitivity specifically in adipose tissue[9]. However, the mechanism by which NE influences metabolic pathways has been unexplained.

Amongst the actions of NE, it can proteolytically cleave corticosteroid binding globulin (CBG; *SERPINA6*)[10]. In humans, CBG binds > 85% of plasma cortisol with high affinity rendering it 'inactive', with only the free unbound fraction (between 5–10%) able to enter target cells and exert transcriptional effects. Inactivating mutations in *SERPINA6* alter the plasma distribution of glucocorticoids, and cause features consistent with an active role for CBG in the delivery of glucocorticoids to target tissues[11–15]. However, evidence that physiological variation in CBG influences glucocorticoid action is lacking. In vitro, NE cleaves CBG at its reactive centre loop, reducing its affinity for glucocorticoids[10]. It has been speculated that NE-mediated CBG cleavage is a mechanism to increase local glucocorticoid exposure at sites of inflammation in vivo[10,16–18]. However, the hypothesis that manipulation of the NE/AAT/CBG axis regulates glucocorticoid tissue exposure has not been tested in vivo.

Here we show that NE deficiency in diet-induced obese mice reduced visceral adipose tissue glucocorticoid levels, an effect that contributes to improved whole body insulin sensitivity. Moreover, humans with reduced AAT levels due to polymorphisms in *SERPINA1* have amplified glucocorticoid exposure in adipose and skeletal muscle, in parallel with an increased circulating free glucocorticoid fraction. These findings establish a physiological role for the NE/AAT/CBG axis in regulating glucocorticoid action that may link inflammation with adverse metabolism.

## Results

### Neutrophil elastase increases glucocorticoid action in mouse and human serum

In vitro studies have previously demonstrated the ability of NE to cleave CBG in human serum samples, but have not tested the effect on glucocorticoid action[10,16,19]. To confirm proteolytic cleavage reduces the capacity of CBG to bind glucocorticoids, in parallel with a hypothesised increase in glucocorticoid action, we assessed the effect of NE/AAT treatment on CBG and glucocorticoids in serum from mice and humans, (Fig. 1). To demonstrate the role of AAT in this axis, we first confirmed that AAT inhibits NE activity in a dose-dependent manner (Fig. 1a). In both mouse and human serum, NE reduced the binding capacity of CBG for corticosterone and cortisol respectively (Fig. 1b, c), an effect that was dose-dependent (Supplementary Fig. 1a). Pre-incubation with AAT blocked the NE-mediated suppression of CBG binding capacity (Fig. 1b, c). In parallel, NE treatment increased free corticosterone and free cortisol in mouse and human serum respectively (Fig. 1d, e), effects that were similarly dose-dependent (Supplementary Fig. 1b). Pre-incubation of NE with AAT prevents the NE-mediated increase in free glucocorticoids in both mouse and human serum (Fig. 1d, e). To determine whether this increase in free glucocorticoids exerted a functional transcriptional effect, we utilised a

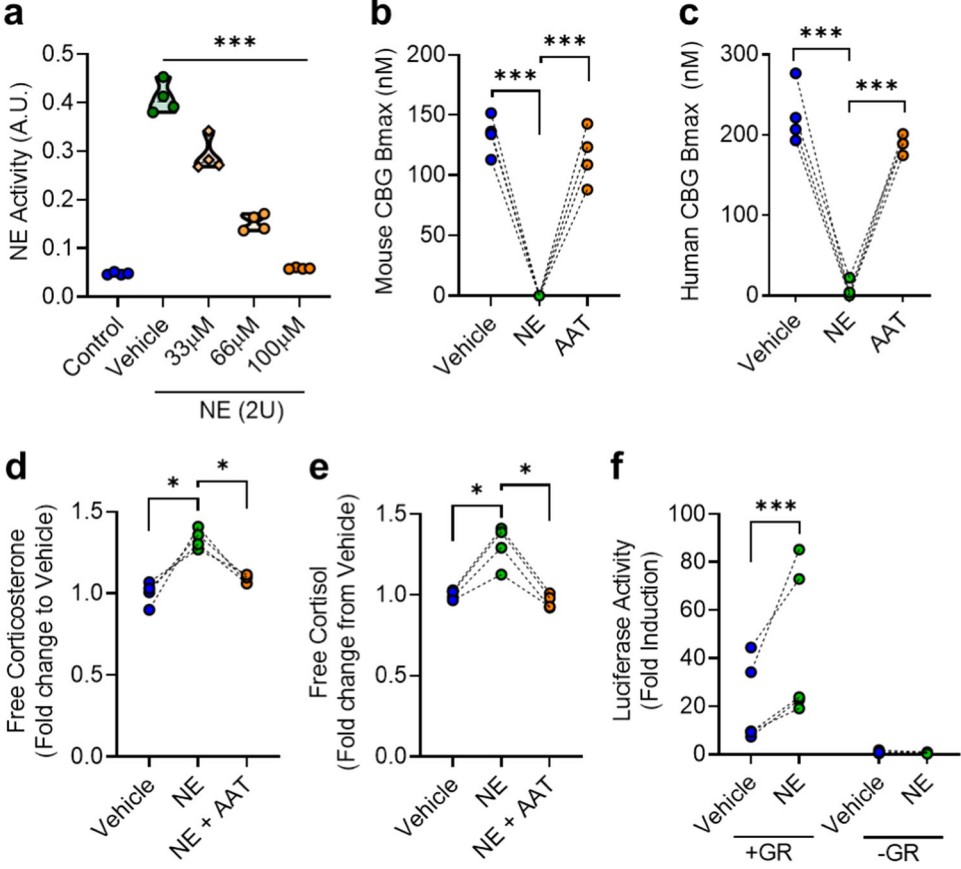

**Fig. 1 | The NE/AAT/CBG axis regulates glucocorticoid action in vitro. a** NE activity is inhibited with indicated concentrations of AAT. Data are presented as mean −/+ SD. (A.U.; arbitrary units). **b**, **c** Treatment of 1:50 diluted mouse serum (**b**) and human serum (**c**) with either NE alone (4 U; green circle) or NE + AAT (100 μM; orange circle) for 10 min at 37 °C. Data are presented as individual points for *n* = 4 biological replicates. **d**, **e** Free glucocorticoid in diluted mouse serum (**d**) and human serum (**e**) treated with either NE alone (4 U; green circle) or NE + AAT (100 μM; orange circle). **f** Glucocorticoid-responsive luciferase reporter activity in human embryonic kidney (HEK293) cells -/+ transfection with human glucocorticoid receptor (GR). Cells were treated for 24 h with 1:50 diluted human serum -/+ NE (4 U, 10 min, 37 °C). Data are presented as individual points for *n* = 4 biological replicates. Data are presented as individual points for *n* = 4 biological replicates. Data analysed by RM one-way ANOVA with Tukey's post-hoc tests (**a**–**e**), or two-tailed paired t-test (**f**). *$P < 0.05$, ***$P < 0.001$.

glucocorticoid receptor (GR)-deficient human cell line (HEK293), and transfected in a glucocorticoid-responsive luciferase reporter (MMTV-luciferase)[20]. In cells co-transfected with GR, but not in controls without GR transfection, incubation with NE-treated serum increased luciferase activity to a greater extent than untreated serum, consistent with increased glucocorticoid action (Fig. 1f). Together these data establish a NE/AAT/CBG axis controlling glucocorticoid action in vitro.

## Neutrophil elastase-deficiency improves diet-induced glucose intolerance and insulin resistance in male but not female mice

To determine the influence of high-fat diet (HFD) on NE, we assessed systemic and tissue NE levels in male and female C57B/6 J mice (Supplementary Fig. 2). Compared to chow-fed controls, 8-week HFD did not increase circulating NE levels in either sex (Supplementary Fig. 2a, e). However, in line with reports of obesity-induced neutrophil infiltration[9,21], HFD administration in male mice increased NE levels in visceral gonadal adipose tissue (gWAT) (Supplementary Fig. 2b), but with no changes observed in either subcutaneous adipose (sWAT) or liver (Supplementary Fig. 2c, d). In contrast, HFD had no effect on either gWAT or liver NE levels in female mice (Supplementary Fig. 2f, g). AAT levels in male mice were not altered in either the circulation or gWAT following HFD (Supplementary Fig. 2h, j respectively), revealing an increased NE:AAT ratio only in gWAT (Supplementary Fig. 2k).

To determine if removal of NE impacted systemic and tissue glucocorticoid action, we first sought to establish the metabolic consequences of NE deficiency in diet-induced obese male and female mice lacking neutrophil elastase (*Elane*[-/-]). While male *Elane*[-/-] mice are reported to be protected from HFD-induced metabolic dysregulation[8], this has not been reported previously in females. We fed male and female mice a high fat, high-sucrose diet (58% kcal fat) for 8 weeks. Male ELA-KO mice were protected from diet-induced metabolic dysregulation. Compared to their wild-type (WT) littermates, male *Elane*[-/-] mice gained less weight (Fig. 2a, b) and displayed improved glucose tolerance (Fig. 2e, f). Fasting insulin and HOMA-IR were significantly reduced (Fig. 2i, k), while the insulin response to glucose stimulation tended to be higher ($P = 0.1$) (Fig. 2j) in male *Elane*[-/-] mice compared with WT controls. To further investigate diet-induced metabolic dysregulation, we measured insulin tolerance and calculated the rate of glucose disposal ($K_{ITT}$). Male *Elane*[-/-] mice demonstrated an improved response to insulin (Fig. 2o) with a significantly enhanced $K_{ITT}$ (Fig. 2p). In contrast, none of these genotype effects were observed in female mice, with NE deficiency having no impact on weight gain (Fig. 2c, d) or glucose and insulin responses (Fig. 2g/h, l–n, q/r).

## Systemic glucocorticoid distribution is not impacted in HFD-fed *Elane*[-/-] mice

As expected, serum NE was undetectable in male and female ELA-KO mice (Supplementary Figs. 3a and 4a). Total and free circulating glucocorticoid concentrations obtained under cull stress conditions were unaltered in male *Elane*[-/-] mice compared to WT controls (Fig. 3a, b), including when calculated as % free corticosterone (Fig. 3c). Moreover, serum CBG binding capacity was no different between male *Elane*[-/-] and WT mice (Fig. 3d). Similarly, female *Elane*[-/-] mice did not display differences in systemic glucocorticoid readouts, with no changes in total or free corticosterone, or in CBG Bmax, compared to WT controls (Supplementary Fig. 4). Together, these data are consistent with a lack of systemic NE action on the circulating glucocorticoid distribution in the presence of AAT.

## HFD-fed male *Elane*[-/-] mice display reduced glucocorticoid action in visceral adipose tissue

While HFD does not increase systemic NE levels in either male or female mice, local action is likely increased in visceral adipose where NE levels are elevated in male, but not female mice (Supplementary

Fig. 2). To determine if tissue NE impacts local tissue glucocorticoid exposure, we investigated corticosterone action in key metabolic tissues including liver and adipose tissue. In male mice, visceral adipose (gWAT) and liver weights were reduced in *Elane*[-/-] mice vs WT controls (Fig. 3e). Serum triglyceride levels were lower in *Elane*[-/-] mice (Fig. 3f). Adipocytes from gWAT were also reduced in size (Fig. 3g, h). Strikingly, this was accompanied by reduced gWAT corticosterone levels compared to WT controls (Fig. 3i), in parallel with significant alterations in several adipose transcripts (Fig. 3j), including reduced glucocorticoid-responsive genes (*Per1*, *Fkbp5* and *Sgk1*) and pro-inflammatory markers (*Tnfa*, *Cd68*, *Itgax*) mRNA, and increased abundance of adiponectin mRNA *(Adipoq)*. Despite reduced adipocyte size, expression of *Pparg* was increased in *Elane*[-/-] mice. No changes were observed in transcripts for the glucocorticoid-regenerating enzyme 11β-HSD1 (*Hsd11b1*) or in genes regulating lipid metabolism (*Pnpla2*, *Lpl*).

In contrast with gWAT, no changes were observed in corticosterone content in sWAT from male *Elane*[-/-] mice (Supplementary Fig. 3d), in keeping with unaltered NE changes observed in this adipose depot under HFD conditions (Supplementary Fig. 2c). There were only minor changes in transcript levels, including reduced *Ccl2* and *Cd68* (Supplementary Fig. 3e).

In liver, histological analysis revealed significantly reduced lipid content in *Elane*[-/-] mice compared to controls (Fig. 3k, l). However, similar to sWAT, no differences were observed in hepatic corticosterone levels between genotypes (Fig. 3m), in line with unaltered NE changes observed under HFD conditions (Supplementary Fig. 2d). Moreover, we did not observe changes in either hepatic gluconeogenic or glucocorticoid-responsive transcripts (Fig. 3n).

## The improved metabolic phenotype of male *Elane*[-/-] mice is adrenal-dependent

Having established a relationship between NE-deficiency and reduced glucocorticoid action in visceral WAT in male mice, we tested whether the metabolic protection of *Elane*[-/-] male mice was dependent on adrenal steroids. We performed bilateral adrenalectomies on *Elane*[-/-] mice and WT littermates, with separate sham-operated controls for each genotype. In line with above observations, in sham-operated adrenal-intact animals *Elane*[-/-] mice gained less weight than their WT littermates on HFD. This protective effect was lost in adrenalectomized animals (Fig. 4a, b). Assessment of glucose tolerance revealed a similar outcome, with *Elane*[-/-] mice lacking endogenous glucocorticoids (and other adrenal steroids) having no further improvement in their glucose tolerance compared to WT littermate controls (Fig. 4c, d). Notably, adrenal-intact *Elane*[-/-] mice had significantly improved glucose tolerance compared to adrenalectomized *Elane*[-/-] mice ($P = 0.027$). The insulin response to glucose stimulation was similar in all groups (Fig. 4e). HOMA-IR analysis indicated that intact *Elane*[-/-] mice had improved insulin sensitivity compared to WT littermates. This difference was not observed in adrenalectomized *Elane*[-/-] mice compared to adrenalectomized WT controls (Fig. 4f). However, adrenal-intact *Elane*[-/-] mice had a significantly lower HOMA-IR index compared to adrenalectomized *Elane*[-/-] mice ($P = 0.041$). These data support an adrenal-dependent mechanism mediating the improved insulin resistance phenotype observed in HFD-fed *Elane*[-/-] male mice.

## Reduced AAT levels in humans increases free cortisol fraction and tissue cortisol exposure

Our murine data showed that NE deficiency is associated with reduced adipose glucocorticoid action, therefore we next set out to investigate the potential translational relevance. In humans, NE deficiency is a rare genetic disorder and so we took an alternative approach in testing the NE/AAT/CBG axis, utilising the relatively common paradigm of AAT deficiency. Here we studied carriers of heterozygous mutations in *SERPINA1* that results in reduced circulating AAT levels

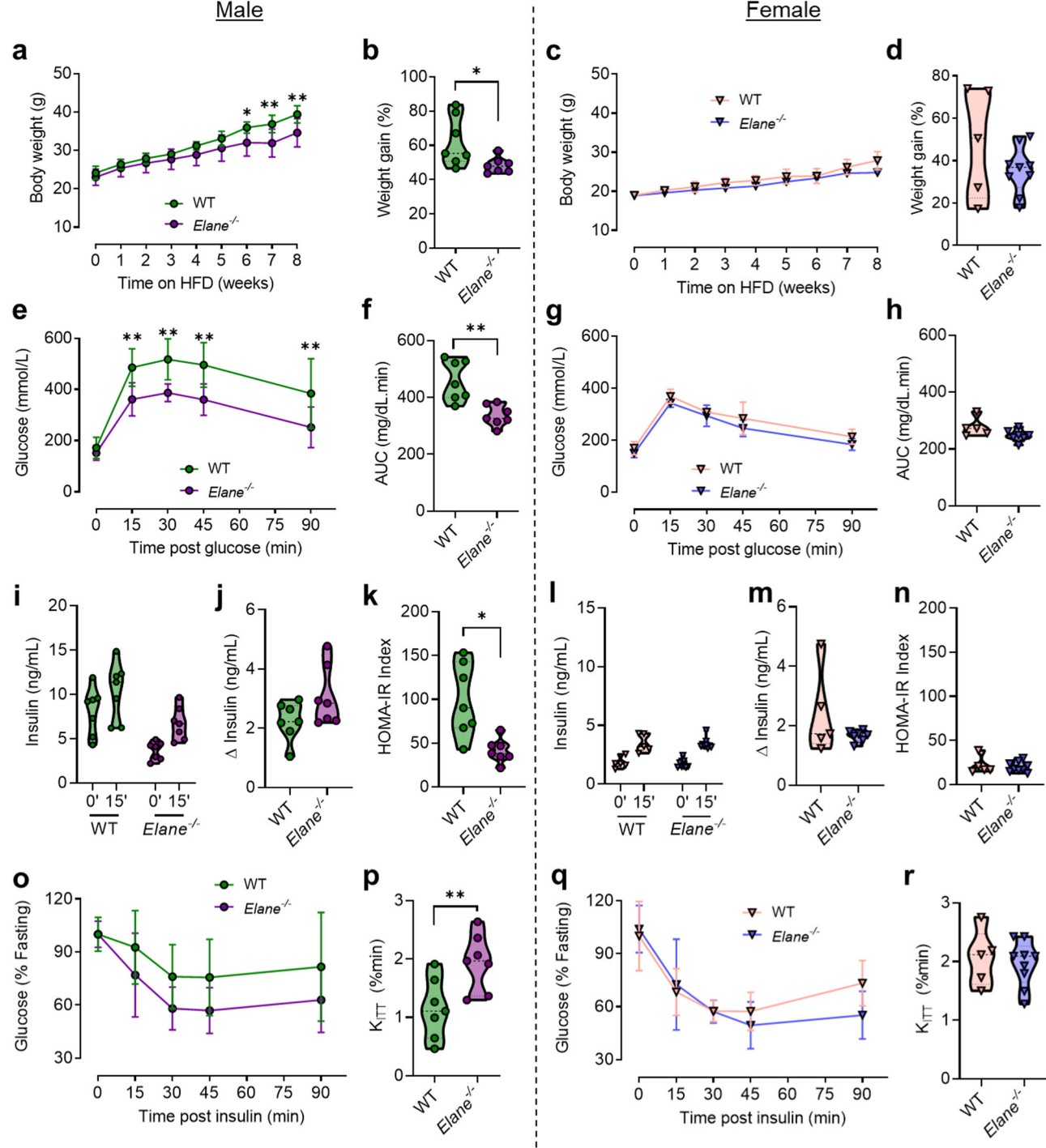

**Fig. 2 | Neutrophil elastase deficiency improves HFD-induced glucose intolerance and insulin resistance in male, but not female mice. a**–**d** Body weight of male and female neutrophil elastase deficient (*Elane*[−/−]; purple circle for male, blue triangle for female) mice or wild-type (WT; green circle for male, ivory triangle for female) littermates fed a high-fat diet for 8 weeks. Corresponding weight gains are depicted. **e**–**h** Blood glucose levels during an IPGTT in mice with the indicated genotypes and sex. Corresponding areas under the curve (AUCs) are depicted. **i**–**n** Plasma insulin levels during 0 min and 15 min time points of IPGTT in mice with

the indicated genotypes and sex. Corresponding changes in plasma insulin post-glucose are depicted. HOMA-IR measure of insulin sensitivity are depicted. **o**–**r** Blood glucose levels during insulin tolerance test (ITT) in mice with the indicated genotypes and sex. Glucose decay rate during the ITT (kITTs) are depicted. Data are presented as mean (dashed line) -/+ SD (dotted line). Male WT *n* = 7, male *Elane*[−/−] *n* = 7, female WT *n* = 5, female *Elane*[−/−] *n* = 9. Data analysed by RM two-way ANOVA with Sidak post-hoc tests (**a, c, e, g, i, l, o, q**), or two-tailed unpaired t-tests (**b, d, f, h, j, k, m, n, p, r**). *$P < 0.05$, **$P < 0.01$.

enhanced neutrophil protease action[22]. Importantly, these subjects are asymptomatic, making them an ideal group to study as they do not present with the liver dysfunction associated with symptomatic AAT deficiency in those with homozygous *SERPINA1* mutations[23]. We recruited 16 asymptomatic *SERPINA1* heterozygous subjects and 16

matched healthy controls. Characteristics of the study participants are outlined in Table 1 and Supplementary Fig. 7 (characteristics are presented graphically by *SERPINA1* allele and by Sex in Supplementary Figs. 8 and 9 respectively), with circulating AAT levels ~30% lower in AAT-deficient heterozygotes.

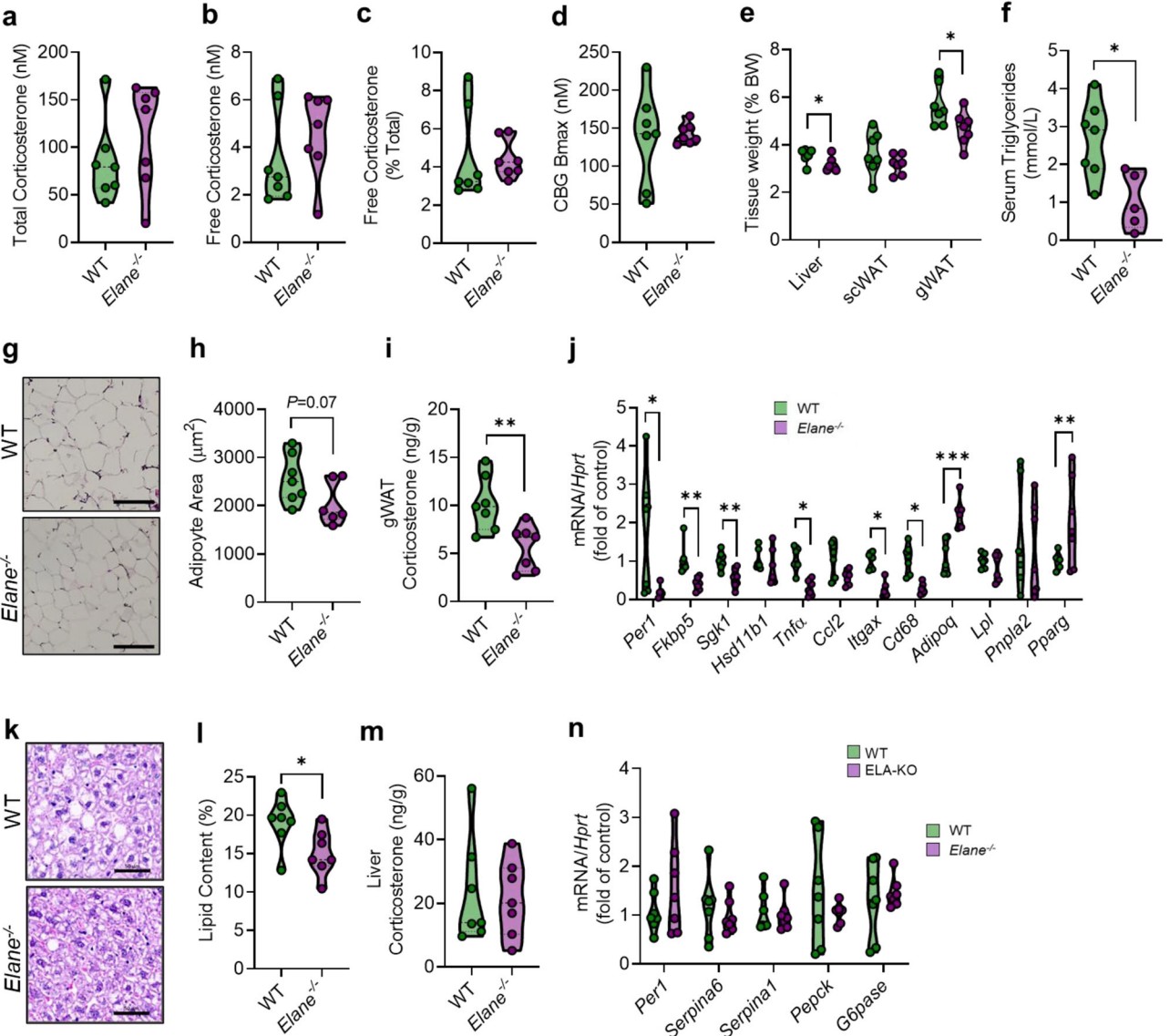

**Fig. 3 | HFD-fed male *Elane*[-/-] mice display reduced glucocorticoid action in visceral adipose tissue. a–d**, Serum glucocorticoid parameters in male *Elane*[-/-] (purple circle) and WT (green circle) mice after 8 weeks of HFD, including total corticosterone (**a**), free corticosterone (**b**), percent free corticosterone (**c**), and CBG binding capacity (**d**). **e** Tissue weights as % body weight for the indicated genotypes. **f**, Serum triglyceride levels (WT *n* = 7, *Elane*[-/-] *n* = 5). **g** Representative H& E images of gonadal adipose (gWAT), scale bar = 100 μm. **h** quantification of adipocyte size for the indicated genotypes. **i**, **j** gWAT corticosterone levels and transcript expression for the indicated genotypes. **k**, **l** Representative H&E images of liver (scale bar = 100 μm) and quantification of lipid accumulation for the indicated genotypes. **m**, **n** Liver corticosterone levels and transcript expression for the indicated genotypes. Data are presented as mean (dashed line) -/+ SD (dotted line). Male WT *n* = 7, male *Elane*[-/-] *n* = 7 unless otherwise stated. Data analysed by two-tailed unpaired t-tests. *$P < 0.05$, **$P < 0.01$, ***$P < 0.001$.

To assess the dynamics of tissue glucocorticoid exposure, we performed arterio-venous (A-V) plasma sampling across tissues including abdominal subcutaneous adipose and skeletal muscle at regular intervals during steady-state 9,11,12,12-[²H]₄-cortisol (D4-cortisol) tracer infusion (Fig. 5). D4-cortisol infusion enables quantification of both endogenous cortisol production (measured by the rate of dilution of D4-cortisol by cortisol) and 9,12,12-[²H]₃-cortisol (D3-cortisol) production[24]. Endogenous cortisol production serves as an index of net cortisol production from all sources, including the adrenal gland, while D3-cortisol production serves as a specific measure of cortisol regeneration by the enzyme 11βHSD1 (Supplementary Fig. 10a). Together, this approach allows quantification of in vivo whole body glucocorticoid appearance and clearance rates, in parallel with glucocorticoid release across tissue.

As expected, arterialised total cortisol declined during the time course of D4-cortisol infusion, with a similar reduction observed in both groups (Fig. 5a). Arterialised D4-cortisol concentrations were in steady state by 180 minutes, when the coefficient of variation for the D4-cortisol:cortisol ratio (Fig. 5b) and the D4-cortisol:D3-cortisol ratio (Fig. 5c) was < 5% in both study groups. Importantly, D4-cortisol clearance (calculated as ratio of infusion rate and concentration) was unchanged over time, and not different between groups (Supplementary Fig. 10c). Arterialised free cortisol levels during steady-state infusion were higher in AAT-deficient heterozygotes (Fig. 5d). However, no changes were observed in total cortisol levels (Fig. 5e) or in CBG binding capacity (Fig. 5f). Regression analyses demonstrated that circulating AAT ($P = 0.0378$) and CBG Bmax ($P = 0.0028$) significantly predicted free cortisol (Supplementary Table 1). Whole body rates of appearance (Ra) of cortisol were not different between groups (Fig. 5g), but D3-cortisol production was reduced (Fig. 5h), indicating reduced whole body 11βHSD1 reductase activity. Circulating data from D4-cortisol

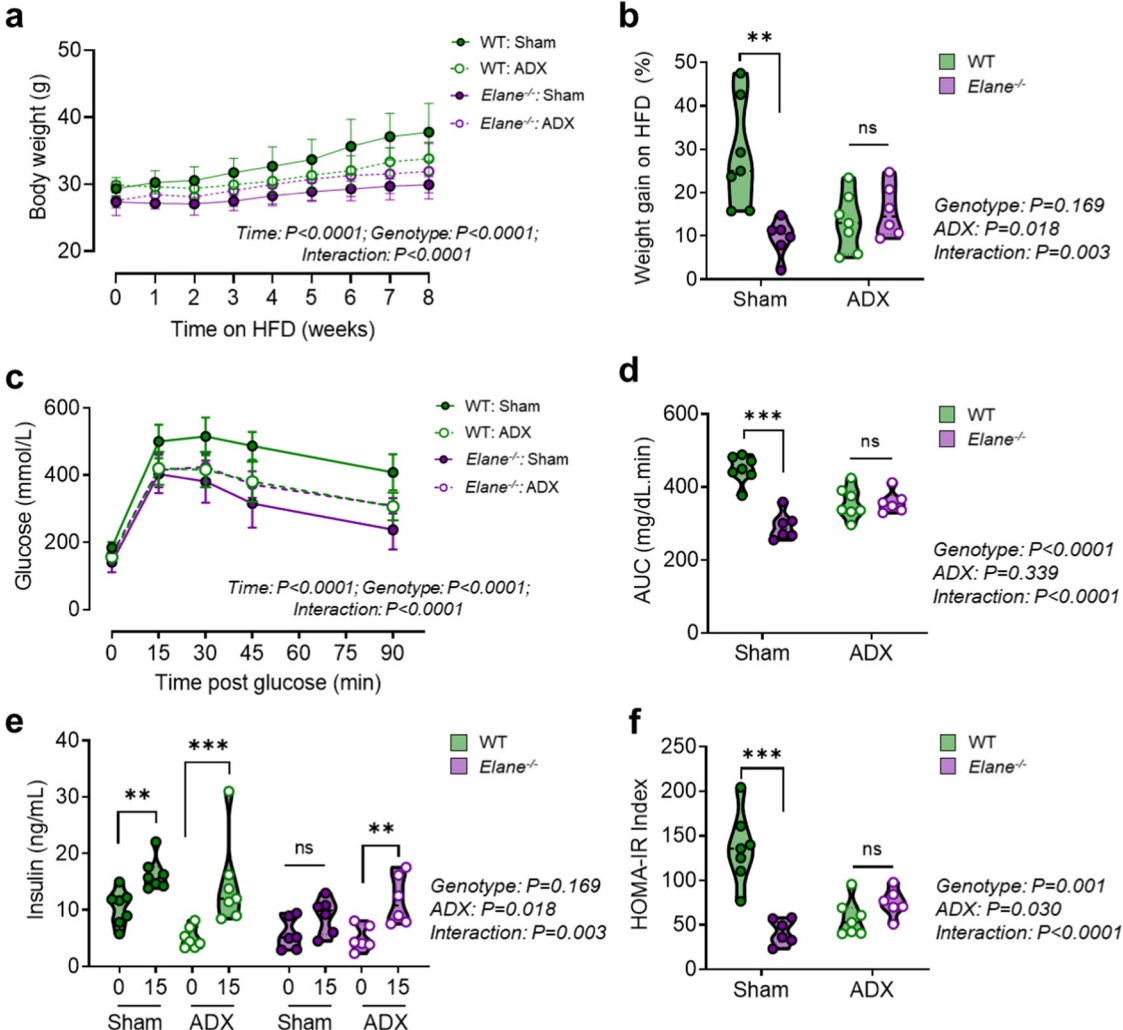

**Fig. 4 | The improved metabolic phenotype of male *Elane⁻/⁻* mice is adrenal-dependent. a, b** Body weight of male neutrophil elastase deficient (*Elane⁻/⁻*; purple symbols) mice or wild-type (WT; green symbols) littermates who underwent bilateral adrenalectomy (ADX; white-filled symbols) or sham-surgery (Sham; colour-filled symbols) prior to 8-week HFD. Corresponding weight gains are depicted. **c, d** Blood glucose levels during an IPGTT in mice with the indicated genotypes and surgery. Corresponding areas under the curve (AUCs) are depicted. **e** Plasma insulin levels during 0 min and 15 min time points of IPGTT in mice with the indicated genotypes and surgery. **f** HOMA-IR measure of insulin sensitivity with the indicated genotypes and surgery. Data are presented as mean (dashed line) -/+ SD (dotted line). WT: Sham *n = 7, Elane⁻/⁻*: Sham *n = 7*, WT: ADX *n = 7, Elane⁻/⁻*: ADX *n = 6*. Data analysed by RM two-way ANOVA (**a, c, e**) or two-way ANOVA with Sidak's post-hoc tests (**b, d, f**). ANOVA results are indicated beside graphs. *$P < 0.05$, **$P < 0.01$, ***$P < 0.001$. ns = not significant.

infusion is also presented by *SERPINA1* allele and by Sex in Supplementary Figs. 11 and 12.

To assess potential release of glucocorticoids as a result of local CBG cleavage across a tissue bed, we determined the net balance of cortisol and D4-cortisol across adipose and skeletal muscle. Unfortunately, technical difficulties arising from blocked cannulae resulted in an inability to measure glucocorticoid dynamics across adipose tissue. However, in skeletal muscle during steady state infusion, cortisol release was significantly increased in AAT-deficient heterozygotes (Fig. 5i). Moreover, D4-cortisol release also tended to be higher ($P = 0.059$) in this group, consistent with release from CBG across skeletal muscle (Fig. 5i). Of note, the rate of appearance of D3-cortisol across the tissue was also elevated (Supplementary Fig. 10b), indicating that while it may not be the primary source of cortisol release, 11βHSD1 is likely contributing to the net release across skeletal muscle. The free cortisol fraction was higher in AAT-deficient heterozygotes in both arterial and venous samples across skeletal muscle, with a trend towards higher free fraction in venous samples (Fig. 5j). Moreover, calculation of net balance revealed greater release of free cortisol across skeletal muscle in AAT-deficient heterozygotes (Fig. 5k),

supporting an inhibitory role for AAT in preventing local CBG cleavage across tissue.

## AAT-deficient heterozygotes do not have altered HPA axis activation

Having established that the circulating free fraction of cortisol is elevated in AAT heterozygotes under exogenous tracer infusion, we sought to confirm our finding in unperturbed 'tracer-independent' plasma and test if there was a loss of central HPA axis feedback. This was achieved using a CRASH (Combined Receptor Antagonist Stimulation of the HPA axis)[25] test, which evaluates endogenous negative feedback of the HPA axis through combined blockade of the two steroid receptors which bind glucocorticoids, namely GR and MR (mineralocorticoid receptor) as part of a randomised, double-blind crossover study, using either a combination of RU486 (GR antagonist) and spironolactone (MR antagonist) or placebo (Supplementary Fig. 13) in the same participants as above.

We first assessed systemic glucocorticoid physiology at baseline in the 'placebo' arm of this study. Confirming our initial finding in 'tracer-infused' plasma, the circulating free cortisol fraction was

**Table 1 | Baseline characteristics of participants with heterozygous mutations in *SERPINA1* and matched controls**

| Baseline characteristics | Control group | AAT[+/-] group | P-Value |
|---|---|---|---|
| Male/Female | 4/12 | 2/14 | 0.65 |
| Age (y) | 56 (31–70) | 59 (40–71) | 0.32 |
| BMI (kg/m²) | 26.3 (20.5–33.3) | 30.4 (20.3–34.0) | 0.21 |
| Fasting glucose (mmol/L) | 5.0 (0.1) | 5.0 (0.1) | 0.88 |
| Fasting insulin (pmol/L) | 51.1 (5.4) | 74.3 (17.4) | 0.33 |
| HOMA-IR | 1.7 (0.2) | 2.5 (0.7) | 0.40 |
| AAT (mg/dL) | 565 (23) | 411 (27) | 0.0002 |
| Neutrophil count (x10⁹/L) | 3.6 (0.3) | 3.8 (0.2) | 0.68 |
| Elastase activity (A.U.) | 0.26 (0.02) | 0.35 (0.03) | 0.014 |
| CBG (nM) | 336.8 (9.8) | 367.2 (20.7) | 0.19 |

Data are number of participants or mean (±SEM) for baseline characteristics of subjects in control and AAT +/- groups (n = 16 per group, with exception of Fasting Insulin and HOMA-IR in which n = 15 for Control group). The sex composition of each group was compared using Fisher's exact test, other data analysed by two-tailed unpaired t test.

significantly elevated in AAT-deficient heterozygotes (Fig. 6a), while total cortisol levels (Fig. 6b) and CBG levels (Fig. 6c) were similar between groups. Consistent with our hypothesis, subcutaneous abdominal adipose biopsies from AAT-deficient heterozygotes displayed significantly increased cortisol levels compared to controls (Fig. 6d), and this was further supported by an increase in the glucocorticoid-responsive transcripts *PER1*, *SGK1*, and *LPL* (Fig. 6e).

As expected, combined GR and MR blockade increased total cortisol levels in the control group (Fig. 6b) but did not reveal any striking differences in HPA axis activation between groups, either in terms of free cortisol (Fig. 6a) or CBG levels (Fig. 6c). This suggests that the impact of lower AAT levels on tissue glucocorticoid exposure does not extend to central feedback sites such as the hypothalamus or pituitary. Similar to the circulating data, CRASH increased adipose cortisol levels, but no differences were observed between AAT-deficient heterozygotes and controls (Fig. 6d).

## Discussion

We provide the first in vivo evidence in mice and humans that manipulation of NE or AAT impacts tissue glucocorticoid exposure. Using complementary approaches to interrogate the NE/AAT/CBG axis, we find that genetic deletion of NE in mice reduces adipose corticosterone action, while people with genetically reduced AAT levels display enhanced adipose and skeletal muscle cortisol exposure (Fig. 7). Our findings provide key insights into the physiological mechanisms that regulate local glucocorticoid action, highlight a significant pathway linking inflammation and metabolism, and underscore the potential therapeutic opportunities to manipulate glucocorticoids and reduce cardiometabolic risk.

One of the key findings from this work is the localised effects of NE/AAT manipulation on glucocorticoid activity. In mice, we used the paradigm of diet-induced obesity and elevated NE to study the impact of removing NE on adipose glucocorticoid exposure. Neutrophil infiltration into obese adipose tissue has been recognised as an important regulator of adipose function and whole-body insulin resistance[7,8,26]. In particular, the role of the serine protease NE has been investigated as a key driver of obesity-induced insulin resistance. While some have reported systemic increases in NE under HFD conditions[8], others have shown a more localised tissue response[9,27]. Indeed, we observed no effect of HFD on circulating NE levels, instead demonstrating a tissue-specific response, with increased NE in

visceral gWAT, but no effect in subcutaneous sWAT or liver. While the underlying reasons for this are unclear, this may be due in part to the different high-fat diets utilised, with a previously used 60% kcal fat diet reported to induce greater insulin resistance in C57BL6/J mice than our 58% kcal fat with sucrose[28]. Consistent with our findings, studies of different adipose depots have previously reported that visceral gWAT is the primary site of neutrophil infiltration and NE protein under obesogenic conditions[9], with little to no neutrophils present in obese subcutaneous sWAT[21]. Importantly, our work in male *Elane*[-/-] mice shows that changes in tissue glucocorticoids under HFD conditions are associated with HFD-induced increases in NE. Reduced corticosterone was only observed in gWAT of *Elane*[-/-] male mice, in parallel with reduced expression of several glucocorticoid-responsive transcripts. In contrast, both sWAT and liver showed no effect in terms of corticosterone levels or glucocorticoid-responsive transcripts. While this does not rule out a role for NE regulation of corticosterone in other tissues, it does suggest that a 'threshold' of NE needs to be reached, most likely to overcome the local inhibitory actions of AAT, in order to engage in proteolytic cleavage of CBG and influence glucocorticoid release. Indeed our assessment of AAT levels in serum and gWAT from male mice further support this theory, with evidence that the ratio of NE:AAT is only increased in gWAT. Of note, reduced corticosterone in gWAT is also associated with a reduction in pro-inflammatory markers. While this appears paradoxical given the established anti-inflammatory properties of glucocorticoids, previous studies have demonstrated that a decrease in glucocorticoid levels does not always correlate with reduced pro-inflammatory gene expression[29–32]. Moreover, it is important to consider that much of the research on glucocorticoid anti-inflammatory action has been carried out with saturating levels of exogenously administered glucocorticoids, which has led to an over-simplified view of their effects. Indeed, accumulating evidence suggests that endogenous glucocorticoids can exert potentiating as opposed to suppressive effects on inflammatory markers[29,31]. In our study, the reduction in pro-inflammatory transcripts most likely reflects a direct consequence of reduced neutrophil elastase.

Notably, this effect of NE deficiency on adipose glucocorticoids does not influence the systemic circulation. Despite circulating NE levels being undetectable in *Elane*[-/-] mice and measurable in WT controls, there was no discernible difference in circulating total and free corticosterone or CBG binding capacity from samples collected under cull stress conditions. Total corticosterone was also assessed at the circadian peak and nadir under basal, unstressed conditions (Supplemental Fig. 3c), but no differences were observed. A limitation here is the inability to quantify free corticosterone in the small volumes obtained from unstressed tail sampling. Given that HFD did not induce systemic NE or AAT levels, this suggests that in WT mice physiological AAT activity is sufficient to inhibit physiological NE action. Indeed, NE deletion did not alter systemic AAT levels (Supplementary Fig. 3b). Notably, our attempts to recapitulate these results obtained in mice with genetic deletion of *Elane* through pharmacological inhibition of NE in mice did not yield the same outcome, either in terms of metabolic protection or adipose corticosterone reduction (Supplementary Fig. 5). However, this is most likely due to the failure of the inhibitor (GW311616A) to suppress NE activity in mice following 8 weeks of administration. Thus, future studies investigating pharmacological manipulation of this axis should confirm NE blockade before commencing investigation.

Consistent with our mouse data, our human study highlights that reduced AAT also results in altered adipose, as well as skeletal muscle, glucocorticoid exposure. Interestingly, our clinical study uncovered some evidence of a systemic effect on glucocorticoid physiology, with an increased circulating free cortisol fraction compared to controls. While this supports our hypothesis of increased NE-mediated CBG cleavage in such people, it was not accompanied by measurable

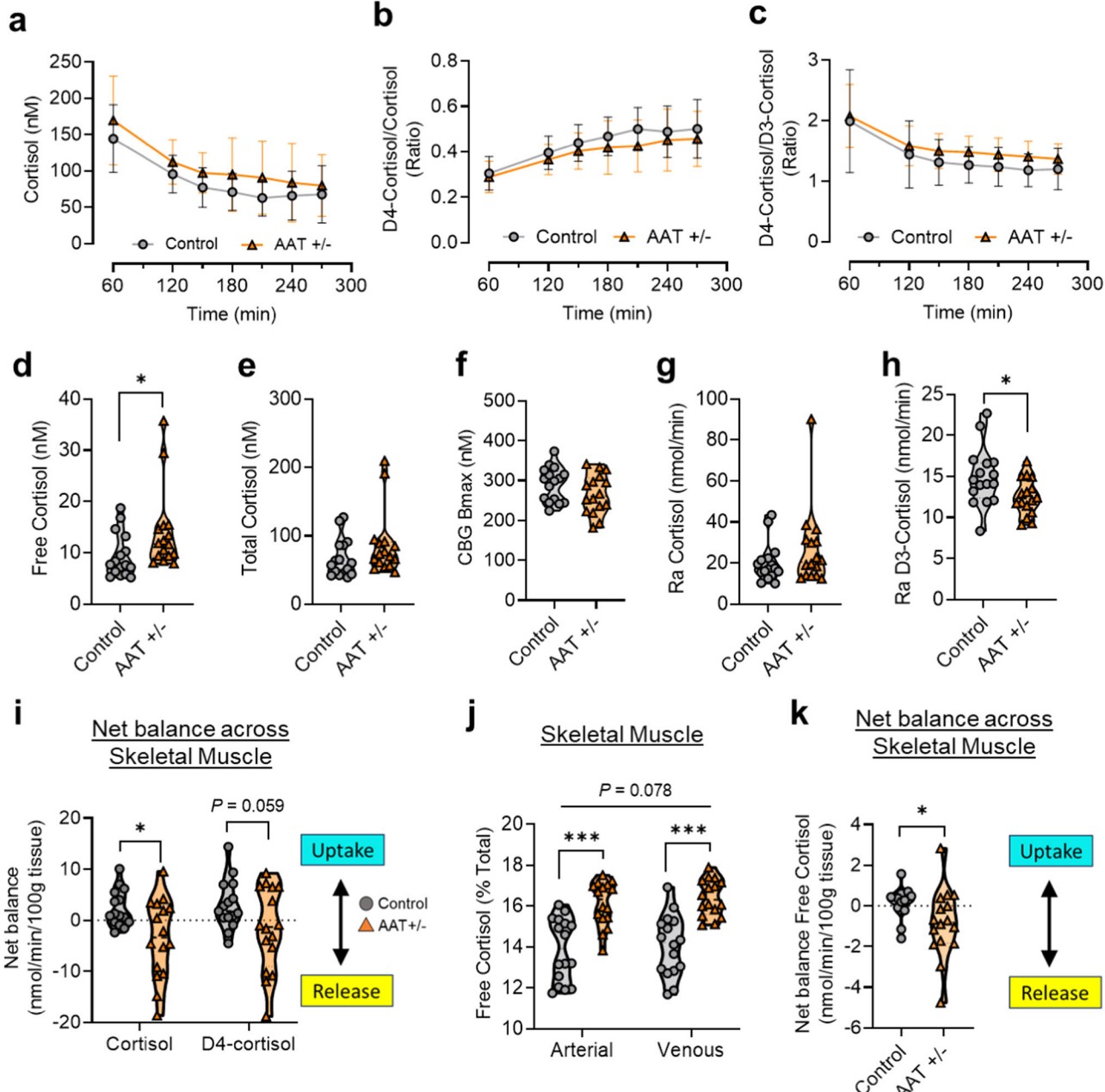

**Fig. 5 | Arterio-venous sampling across tissue reveals increased local glucocorticoid exposure in AAT heterozygotes.** Arterio-venous sampling in subjects with heterozygous mutations in *SERPINA1* (AAT +/-; orange triangles) and matched controls (Control; grey circles) with steady-state 9,11,12,12-[²H]₄-cortisol (D4-cortisol) tracer infusion. **a–c** Plasma profile during D4-cortisol infusion, including endogenous cortisol levels (**a**), D4-cortisol/cortisol ratio (**b**), and D4-cortisol/D3-cortisol ratio (**c**). **d–f** Plasma glucocorticoid profile during steady state (180 – 270 min), including free cortisol (**d**), total cortisol (**e**), and CBG binding capacity (**f**). **g–h** Whole body rate of appearance (Ra) of cortisol (**g**) and D3-cortisol (**h**) during steady state. **i** Net balance of cortisol and D4-cortisol across skeletal muscle. **j** Percent free cortisol across skeletal muscle. **k** Net balance of free cortisol across skeletal muscle. Data are presented as mean (dashed line) -/+ SD (dotted line). Control $n = 16$, AAT + / - $n = 16$. Data analysed by Mixed-effects model (**a–e**), two-tailed unpaired t-tests (**d–h, k**), or RM two-way ANOVA with Sidak's post-hoc tests (**i, j**). *$P < 0.05$, ***$P < 0.001$.

changes in CBG. Intriguingly, this increased systemic free cortisol was not 'corrected' by the HPA axis, leading us to hypothesise that there was a potential loss of central negative feedback, but this was not confirmed using combined receptor antagonist stimulation of the HPA axis. It is possible that mild AAT deficiency with subclinical pathology results in increased central drive to the HPA axis. An important caveat is that our measurements are made in venous blood from the hand, albeit arterialised by hand-warming, which may not be representative of capillary blood and interstitial fluid at HPA axis feedback sites (hypothalamus and pituitary).

In humans it is feasible to measure release of cortisol into tissues in vivo. Unfortunately, our attempts to investigate the cortisol balance across adipose were hampered by technical issues. Indeed the success rate of adipose vein cannulation is often as low as two-thirds[33,34], and in this study was < 50%, meaning we were significantly underpowered to perform meaningful analysis. Nonetheless, our data across skeletal muscle are consistent with the hypothesis of local tissue-mediated control of glucocorticoid exposure via the NE/AAT/CBG axis. Our study of AAT-deficient heterozygotes involved recruitment of subjects with either the 'Z' or 'S' *SERPINA1* allele. To

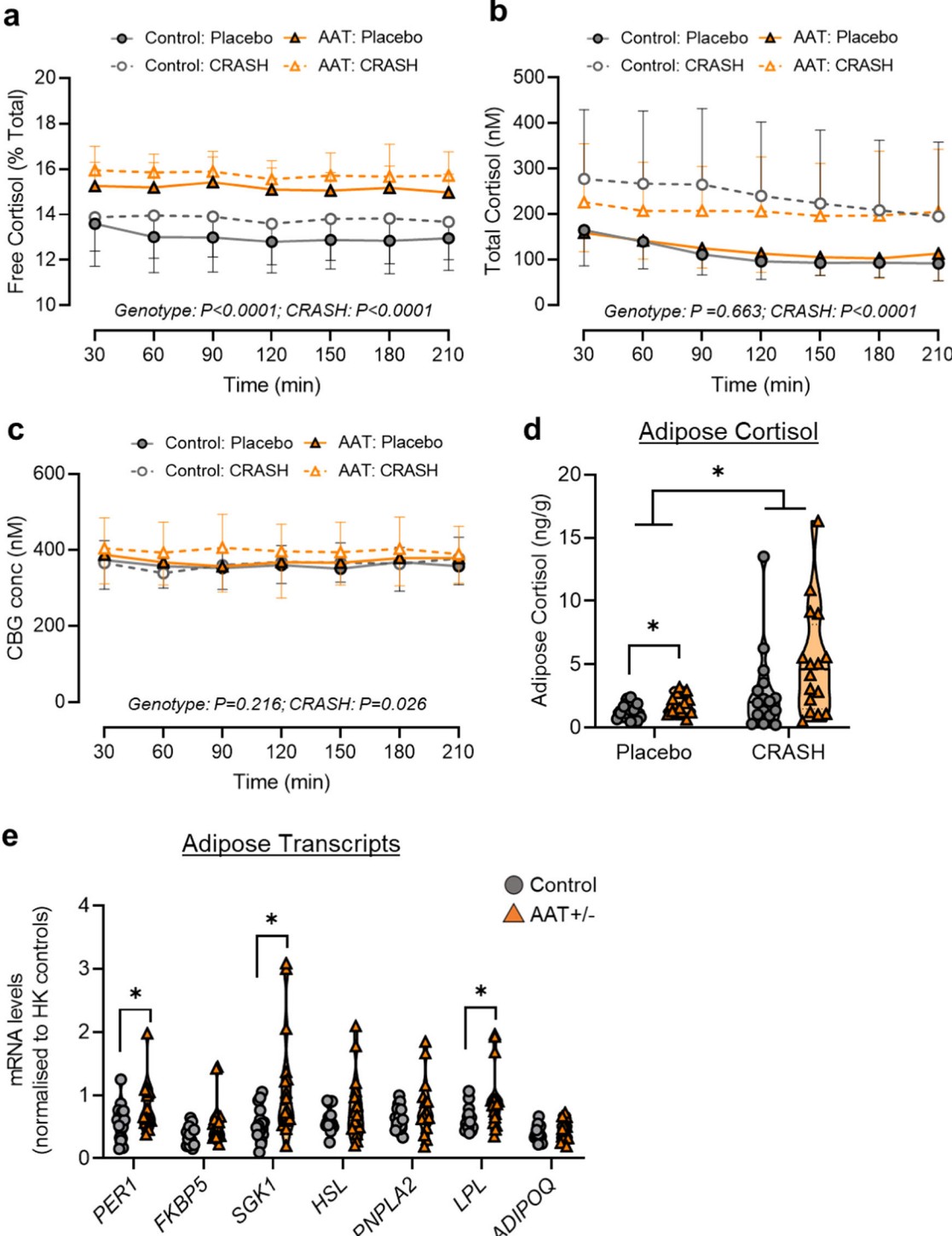

**Fig. 6 | Humans with reduced AAT levels have an increased free cortisol fraction in parallel with increased adipose cortisol exposure. a–c** Circulating glucocorticoid profile in subjects with heterozygous mutations in *SERPINA1* (AAT +/-; orange symbols) and matched controls (Control; grey symbols), as part of a randomised, double-blind crossover study, using either a combination of RU486 and spironolactone (CRASH; white-filled symbols) or placebo (colour-filled symbols), including percent free cortisol (**a**), total cortisol (**b**), and CBG (**c**). **d** Cortisol levels in subcutaneous abdominal adipose from subjects with the indicated genotype and treatment. **e** Transcript expression in subcutaneous abdominal adipose from subjects with the indicated genotype in the placebo group. Data are presented as mean (dashed line) -/+ SD (dotted line). Control: Placebo *n = 16*, AAT + /-: Placebo *n = 16*, Control: CRASH *n = 16*, AAT + /-: CRASH *n = 16*. Data are analysed by Mixed-effects model (effects and corresponding *P*-values indicated on graph) (**a**–**c**), RM two-way ANOVA with Sidak's post-hoc tests (**d**), or two-tailed unpaired t-tests (**e**). *$P < 0.05$.

determine if this heterogeneity accounted for meaningful effects, we performed additional analyses of the subjects at baseline (Supplementary Fig. 8) and during both our tracer study (Supplementary Fig. 11) and our HPA axis 'CRASH' study (Supplementary Fig. 14). While our study was not powered to detect differences between AAT heterozygote groups, it is interesting to note that increased circulating free cortisol is evident in the group carrying the 'Z' allele alone, which has relatively lower AAT levels. No other significant observations were

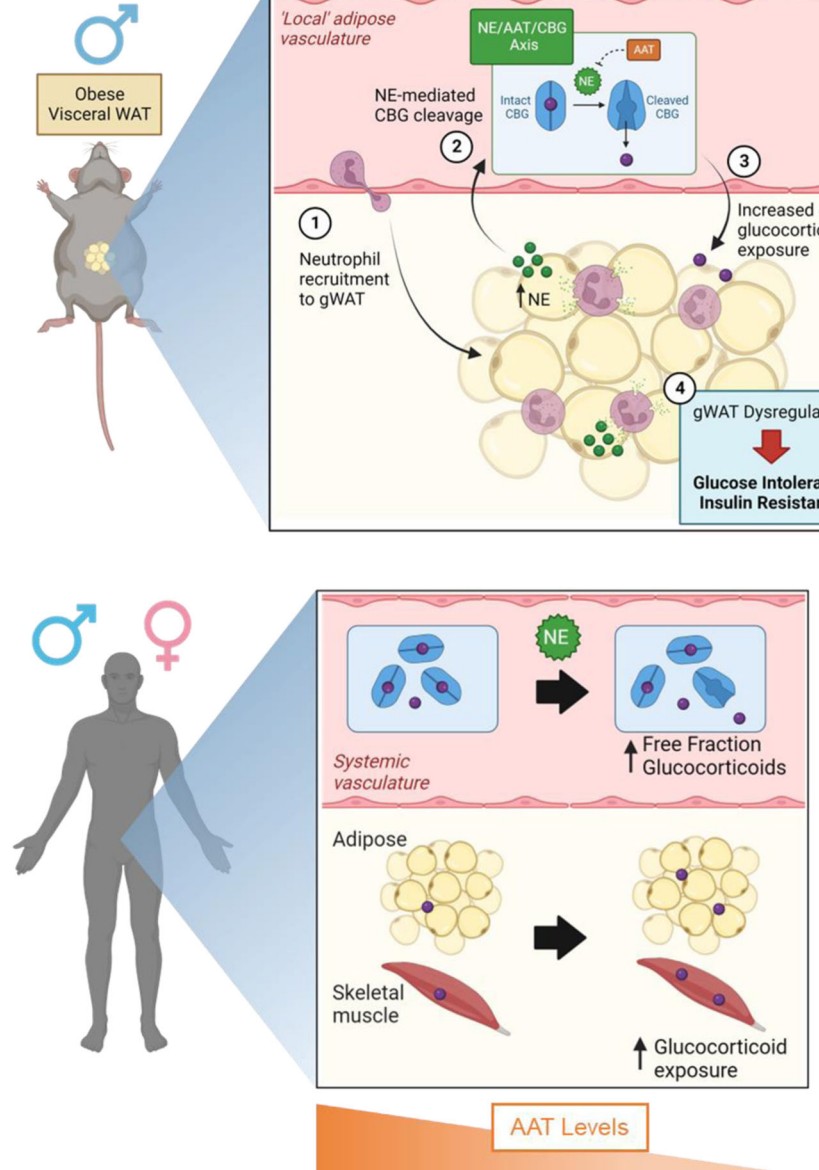

**Fig. 7 | Overview of proposed NE/AAT/CBG control of local and systemic glucocorticoid action in mice and humans.** Illustration of proposed NE/AAT/CBG mechanism of action in murine obesity and in humans with alterations in AAT levels. Created in BioRender (https://BioRender.com/b20b142).

identified when assessing genotype-specific outcomes. Notably, in line with our data, interrogation of UK Biobank data indicates that the SNPs relevant to the AAT heterozygotes do not associate with readouts of obesity or overt metabolic syndrome, although the 'Z' allele is associated with increased trunk mass and trunk fat mass. Together, this suggests these subjects are not predisposed to metabolic syndrome. Of course, any effect of AAT deficiency on obesity or metabolic disease in patients may be confounded by their increased propensity to liver and respiratory disease, both of which are associated with weight loss. A further consideration is that there is likely compensation for the altered glucocorticoid profile we observe in these subjects. For example our tracer study revealed reduced whole-body 11βHSD1 activity, indicating reduced extra-adrenal cortisol generation (most likely in the liver which is a key metabolic tissue). This compensatory mechanism would potentially limit the adverse cardiometabolic effects caused by increased adipose glucocorticoid exposure, making associations at a population-wide level difficult to determine.

Our findings in male mice are congruent with previous reports showing genetic ablation of NE improves obesity-induced insulin resistance[7,8]. We report for the first time that insulin sensitivity is not improved in female *Elane*[-/-] compared to WT controls. Interestingly, our analysis of gWAT from HFD-fed female mice indicates that, in contrast to males, corticosterone is not different in visceral adipose between *Elane*[-/-] and WT mice (Supplementary Fig. 4g). This could be interpreted as evidence supporting the protective metabolic phenotype observed in male *Elane*[-/-] mice being mediated by reduced gWAT corticosterone. However, there is well established sexual dimorphism that protects female mice from obesity-induced metabolic dysregulation[35–40]. Indeed, in our study, we found that in contrast to male mice on HFD, development of glucose intolerance and insulin resistance was less pronounced in female mice, in parallel with no increase in visceral adipose NE levels. It remains a limitation of our study that clear sexual dimorphism in relation to the NE/AAT/CBG axis cannot be stated given the reduced metabolic disturbance in female mice, thus further work is required to delineate the role of neutrophils

in obesity-induced metabolic dysregulation and unpick the sex-specific immune responses to high-fat diet. Although our human study was not powered to demonstrate sex-specific differences, we performed exploratory analyses of the subjects at baseline (Supplementary Fig. 9) and during both our tracer study (Supplementary Fig. 12) and our HPA axis 'CRASH' study (Supplementary Fig. 15) to test if the effect of reduced AAT on glucocorticoid bioavailability was similarly more pronounced in males than females. Perhaps unsurprisingly given the low numbers, we found no discernible sex differences in systemic or tissue parameters. However, of note, our main findings of elevated circulating free cortisol and increased release across tissue hold true even when males are removed. This human data would appear discrepant with our mouse study in which female mice did not respond to NE manipulation, suggesting potential species-specific NE/AAT interactions in the circulation vs local tissues. Nonetheless, future studies investigating the NE/AAT/CBG axis would benefit from study of both sexes, especially given evidence of oestrogen driven differences in CBG levels[41].

Adrenalectomy experiments supported the metabolic protection in male *Elane*[-/-] mice being dependent on glucocorticoids, albeit that we cannot rule out a contribution from other adrenal hormones. Adrenalectomy is known to reduce weight gain and improve metabolic parameters in genetically obese *ob/ob* mice[42], and this parallels our findings in WT mice. However, in our hands male adrenalectomized mice on HFD displayed significantly higher glucose and insulin responses compared to corresponding female mice, indicating that the lack of metabolic improvement with NE deficiency in adrenalectomised mice was not due to lack of glucose intolerance or insulin resistance. Indeed, both glucose intolerance and insulin resistance (HOMA-IR) were greater in adrenalectomized *Elane*[-/-] mice vs adrenal-intact *Elane*[-/-] mice. While this implicates reduced glucocorticoid action as the driver behind the protective effects of NE-deficiency, we cannot rule out the contribution of other adrenal hormones.

A limitation of our studies is that it has not been possible to demonstrate the impact of NE or AAT deficiency on readouts of CBG levels or binding capacity in vivo. While a previous study attempted to address if AAT deficiency in humans impacts NE-mediated CBG cleavage, it was confounded by a number of limitations[43]. Although concluding that CBG cleavage is paradoxically reduced in such patients, the study utilised an immunoassay approach to infer 'cleaved CBG' which has since been invalidated[17], as well as the inclusion of subjects with severe AAT-deficiency who were receiving glucocorticoid therapy. CBG cleavage by NE has previously been demonstrated in serum via gel electrophoresis[10,44,45], however accurate quantification requires affinity purification of CBG from pooled serum samples. Where possible, future studies would benefit from concomitant analyses of CBG readouts. At a tissue level, identifying CBG cleavage is technically even more challenging. While CBG binding capacity has been assayed in cerebrospinal fluid from humans and interstitial fluid from rats[46], obtaining the necessary volumes of interstitial fluid from murine tissues remains a limiting factor. Attempts at measuring CBG binding capacity from whole mouse tissue require extensive perfusion to accurately quantify readouts without 'contamination' from the circulation, and interpretation can suffer from the potential confounding impact of anaesthesia-induced HPA axis activation[47–49]. Nonetheless, it remains a limitation of this and other studies that tissue CBG binding capacity cannot yet be accurately determined. However, our readouts in mice and humans highlight that circulating CBG binding capacity is unaffected by either NE-deficiency or reduced AAT. This is supported by the lack of studies identifying evidence of CBG cleavage in the circulation. Of note, while CBG Bmax values between the two groups in our human study (AAT +/- subjects and controls) are not statistically different, free cortisol levels are significantly correlated with CBG Bmax by regression analysis (Supplementary Table 1). It has long been thought that NE-mediated CBG cleavage does not occur in the

systemic circulation due to the presence of saturating levels of AAT[10]; AAT is produced by the liver and its concentration in the blood is several orders of magnitude higher than that of NE, therefore rapidly neutralising low circulating levels of NE[50]. Together, this suggests that future studies investigating CBG control of tissue glucocorticoid exposure would benefit from advancing methodologies that facilitate labelling of CBG and enable differentiation of intact and cleaved forms within tissues.

While CBG has been identified as being regulated by metabolic factors such as obesity and insulin secretion[51,52], there is little evidence from humans with CBG null mutations or from mice lacking CBG that deficiency alters cardiometabolic risk. The single mouse study investigating diet-induced obesity found that CBG-deficient mice had redistribution of body fat from subcutaneous to visceral depots when placed on HFD compared to control mice, but assessment of other metabolic parameters such as glucose or insulin were not performed[53]. We have considered whether NE could influence tissue glucocorticoid exposure independently of CBG cleavage. A recent report found that an obesogenic diet administered to C57Bl6J mice resulted in increased vascular permeability in visceral gonadal adipose tissue, an effect that was blocked in NE-deficient mice[54]. By enhancing access of CBG-bound glucocorticoids to the interstitial cellular environment with adipose, this could also conceivably increase adipose glucocorticoid action. However this does not take into account that the glucocorticoid is still biologically 'inactive' in its CBG-bound form. As new sampling technologies emerge, further investigation is required to interrogate both CBG physiology and glucocorticoid exposure not only in metabolic tissues such as adipose and skeletal muscle, but to sites of acute inflammation.

In conclusion, these studies provide compelling evidence that both NE and AAT influence glucocorticoid action in mice and humans, respectively. The most parsimonious explanation is that they interact to influence CBG affinity for glucocorticoids locally at sites of inflammation. Together, this reveals a critical axis linking regulation of glucocorticoid action with its effects on inflammation and metabolism.

## Methods

### In vitro studies
Human serum was taken from healthy volunteers in compliance with all relevant ethical regulations and following approval by the Academic and Clinical Central Office for Research and Development (ACCORD) Medical Research Ethics Committee (AMREC) (reference 20-HV-069). Written informed consent was obtained from all participants. Mouse serum was collected from adult (8-12 week) male C57Bl6/JCrl mice bred in-house at the University of Edinburgh. Serum was first diluted in 1:50 in 0.1 M Tris HCL, followed by incubation with 1-4 U of human NE (Elastin products; #SE563) or Tris HCl 0.1 M (Vehicle) for 10 min at 37 °C. In a separate experiment, NE (4 U) was incubated with alpha-1 antitrypsin (100 μM; Sigma; #A9024) for 10 min at 37 °C prior to treatment of serum. Serum CBG binding capacity was measured using an established ligand-saturation assay as outlined below.

### Glucocorticoid-responsive reporter assay
HEK293 cells (ECACC) were cultured in DMEM (Thermo Fisher) supplemented with FBS (10 % v/v; Thermo Fisher), penicillin (100 IU/mL), and streptomycin (100 IU/mL). Cells were seeded at $1 \times 10^5$/ well on a 96-well plate. Following overnight incubation, medium was replaced with Opti-MEM (Lonza) and cells transiently transfected using Lipofectamine 3000 (Invitrogen) with 48 ng pMMTV-LTR-Luc, 48 ng pKC275 (encoding β-galactosidase as internal control) and 4 ng of human GR[20]. After overnight incubation, medium was replaced with serum (1:50 diluted in OptiMEM) pre-treated with neutrophil elastase (4 U) for 10 min at 37 °C or vehicle (0.1 M Tris HCl). 24 h later, luciferase and β-galactosidase activities were measured in cell lysates as previously described[20]. All transfections were carried out in triplicate and

the mean ratio of luciferase/ β-galactosidase activities was calculated. Plasmids were a kind gift from K.E. Chapman, Centre for Cardiovascular Science, University of Edinburgh.

## Serum free glucocorticoids

1,2,6,7 [$^3$H$_4$]-Corticosterone or 1,2,6,7 [$^3$H$_4$]-cortisol (25uCi/mL, Perkin Elmer) was added to mouse or human serum respectively. Samples were diluted 1:50 in Tris HCl 0.1 M (pH 7.5) in a final volume of 100 μL and incubated for 1 h at 37 °C. Samples were treated with 10 μL NE (4 U; Elastin products; #SE563) or with 10 μL Tris HCl 0.1 M (Vehicle) and incubated for 10 min at 37 °C. Retentate (10 μL) was removed and placed in 2 mL scintillation fluid (Ultima Gold, Perkin Elmer). The remainder was loaded onto an ultrafiltration column (Millipore Centrifree Ultrafiltration Device; #4104) and spun at 2000 g for 30 min at 37 °C. Filtrate (10 μL) was removed and added to 2 mL scintillation fluid. Both retentate and filtrate samples were analysed on a scintillation spectrophotometer. Free fraction percentage was calculated as the ratio of cpm filtrate (free glucocorticoid)/ cpm retentate (bound glucocorticoid). In a separate experiment, NE was incubated with AAT (100 μM; Sigma; #A9024) for 10 min at 37 °C prior to treatment of serum.

## Animals

Studies in mice were done in compliance with all relevant ethical regulations under project licences granted by the UK Home Office and were approved by the University of Edinburgh Animal Welfare and Ethical Review Board.

NE-knockout (*Elane*$^{-/-}$) mice (Jackson Laboratory Strain #006112) were bred in-house through heterozygous crosses and housed at 21 °C on a 12 h light/dark cycle (0700–1900) with free access to water and food, as indicated. Male and female C57Bl/6JCrl mice (8 weeks old) were fed either a high-fat diet (58 % kcal fat with sucrose; Research Diets D12331) or chow diet (2.71% kcal from fat; Special Diet ServicesRM1(E) 801002) for up to 8 weeks. Male and female ELA-KO mice and wild-type (WT) littermates (8-10 weeks old) were fed a high-fat diet (58 % kcal fat with sucrose; Research Diets D12331) for up to 10 weeks. A cohort of mice were bilaterally adrenalectomized under isoflurane anaesthesia. Sham-operated were subjected to similar surgery, with adrenal exposed but not removed. Animals were allowed to recover for 7 days before starting high-fat diet. Mice were culled between 0800 h and 1000 h by decapitation to minimise the stress response ( < 1 min between handling and decapitation). Trunk blood was collected and allowed to clot, before being subjected to centrifugation (1500 g, 10 min) to obtain serum.

## Metabolic measurements

Glucose tolerance test (GTT) was performed on mice fasted for 6 h. Animals received an intraperitoneal (IP) injection of glucose (2 g/kg of body weight). Blood was drawn from the tail vein. Blood glucose levels were determined via tail venesection at various time points post-injection using a point-of-care glucometer (Accu-Chek; Roche). A similar protocol was used for the insulin tolerance test (ITT) (injection of 0.75 IU/kg of body weight). Blood was collected from fasted mice and 15 min post-glucose injection to enable assessment of insulin responses to glucose challenge. Plasma was obtained following centrifugation (10,000 g, 5 min), and samples assessed using a commercial insulin ELISA (Millipore; #EZRMI-13K) following manufacturer's instructions. Homoeostatic model assessment of insulin resistance (HOMA-IR) was calculated by using the formula (Fasting insulin (mg/dL) × Fasting glucose (mmol/L)/ 22.5. Serum NE and AAT were quantified using commercial ELISAs (Abcam; NE, ab252356; AAT, ab205088). Diurnal sampling of blood for circadian glucocorticoid quantification was performed by tail venesection (0700 h and 1900h) and collected in EDTA-coated capillary blood tubes (Microvette), prior to analysis of steroids in plasma by liquid chromatography tandem mass-spectrometry (LC-MS/MS).

## Histological analysis

Adipose and liver tissue samples were fixed overnight in 4% paraformaldehyde, before cryoprotection through increasing sucrose gradients (10 %, 20 %, 30 % w/v in PBS) and embedding in OCT. Sections (5 μm) were placed onto slides (SuperFrost, Fisher) and stored at −20 °C until analysis. Sections were stained with hematoxylin and eosin (H&E) and images captured using an Nikon Eclipse Ci-L Plus microscope (Nikon). Quantification of adipocyte area (adipose tissue) and lipid content (liver) were performed using FIJI software (ImageJ) (Adiposoft plugin used for adipocyte analysis). IHC for Ly6G (neutrophil-specific marker) was performed on the BOND RX system (Leica Biosystems) using the BOND Polymer Refine Detection Kit (DS9800, Leica Biosystems). Briefly antigen retrieval was performed using BOND epitope retrieval solution 1 for 20 min, followed by peroxide block for 5 min. Slides were incubated with primary antibody (Ly6G 1:300, mAb #87048, Cell Signalling) for 60 min. Stained slides were imaged using a Nikon Eclipse Ci-L Plus microscope (Nikon). Quantification of DAB of was performed using FIJI software (ImageJ), with parameters for threshold, hue and saturation kept constant across all samples. One sample was excluded from analysis due to a processing error.

## Humans

All human studies were done in compliance with all relevant ethical regulations; following approval by the Academic and Clinical Central Office for Research and Development (ACCORD) Medical Research Ethics Committee (AMREC) (reference 17-HV-032) and by NHS Lothian Research and Development (2017/0193). All participants provided written informed consent prior to any study procedures.

**Participants.** 16 asymptomatic carriers (cases) of AAT deficiency alleles (genotypes PiMS (7 subjects) & PiMZ (9 subjects) in a 'first come' approach) and 16 non-carriers (controls, genotype PiMM) were recruited to this age, sex- and BMI-matched case-control healthy volunteer study. Informed consent was obtained from each participant. All participants completed a comprehensive screening evaluation that included a medical history and physical examination, and standard blood tests to determine eligibility. The following inclusion criteria were required: aged 18-70 years, asymptomatic carriers and non-carriers of AAT deficiency (age-, sex- and body mass index-matched controls), women of a childbearing potential who were willing to use a barrier method of contraception. Potential participants with abnormal screening bloods (full blood count and renal, liver and thyroid function) of clinical significance, those with active acute or chronic medical conditions requiring a therapy (including hormonal contraceptive use), those with history of oral, topical or inhalational corticosteroid use in the preceding six months, or those that were pregnant, seeking to become pregnant or lactating were excluded.

**Study protocol: tracer infusion and arterio-venous sampling.** 9,11,12,12-[$^2$H]$_4$-cortisol (D4-cortisol) was obtained from the Cambridge Isotope Laboratories and dissolved in pharmaceutical-grade ethanol/ water [90:10 (v/v)] and filtered to form sterile stock solutions by Tayside Pharmaceuticals. Participants attended the clinical research facility at 8:00 a.m. after an overnight fast from 11:00 p.m. An anterograde 22 G cannula was positioned in the right antecubital fossa for blood sampling. A retrograde 20 G intravenous cannula was inserted in the deep branch of the medial cubital vein in the antecubital fossa of the left arm for forearm skeletal muscle sampling. A further retrograde cannula was inserted in the dorsum of the left hand. The left hand was placed in a hot box (manufactured in house) heated to 60 °C for 5 min prior to sampling in order to obtain arterialised samples. This technique has been shown to mimic arterial blood and has been used in

previous clinical studies to avoid the need for invasive arterial cannulation[55]. Sampling cannulae were kept patent with a slow infusion of 0.9 % saline. Infusions were stopped and the 0.5 mL dead space was discarded before blood samples were obtained. At $t = -5$ min, a priming dose (0.92 µmol D4-cortisol) was administered for 4 min, followed by steady-state infusion starting a $t = 0$ min (17.2 nmol/min D4-cortisol) for 295 min. Blood samples were obtained at 30 min intervals between $t = 60$ min and $t = 270$ min in potassium EDTA tubes (2.7 mL) prechilled on wet ice. Tubes were centrifuged at 4 °C within 30 min of sampling and stored at −80 °C.

**Study protocol: HPA axis and adipose sampling.** Adipose and systemic glucocorticoid physiology was assessed with Combined Receptor Antagonist Stimulation of the HPA axis (CRASH) testing using spironolactone and RU486 or placebo (Mattsson et al. [25]) in a double-blind randomised crossover design (Supplementary Fig. 13a). Participants attended for study visits on two separate days, separated by at least one week, and with a minimum three-week gap after the arterial-venous study above. On visit days (Supplementary Fig. 13b), participants took either placebo or spironolactone 200 mg/mifepristone 400 mg (Tayside Pharmaceuticals). Participants were given placebo and spironolactone/mifepristone in random order (undertaken by Tayside Pharmaceuticals). Participants were asked to abstain from alcohol the night before and to avoid caffeine after breakfast. Participants were provided written instructions; after taking their capsules at 11:00 a.m., they were advised to have a brunch between 11:00 a.m. and 12:00 p.m. (mifepristone and spironolactone ideally taken with food) and then to fast from noon. Participants presented to the clinical research facility at 3:30 pm., and following a blood pressure check, one anterograde 22 G intravenous cannula was inserted in either arm for blood sampling. Sampling cannula was kept patent with a slow infusion of 0.9% saline. The infusion was stopped and a dead space discarded before blood samples were obtained. A second dose of the capsules the participant had at 11:00 a.m. was ingested by the participant at precisely 4:00 p.m. At $t = 215$ min after 4:00 pm a needle aspiration biopsy of subcutaneous abdominal adipose tissue was obtained as previously described[56] and stored at −80 °C. At $t = 230$ min the sampling cannula was removed and the participant was discharged.

### Metabolic analysis
Fasting glucose and insulin were assayed by Clinical Chemistry in the NHS Lothian laboratory using an Architect c16000 analyser (Abbott Diagnostics). Serum AAT was quantified using a commercial insulin ELISA (Immunology Consultants Laboratory; #E-80A1T) following manufacturer's instructions. Serum elastase activity was measured as described using N-methoxysuccinyl-Ala-Ala-Pro-Val p-nitroanilide, a highly specific synthetic substrate for NE[57]. Serum CBG concentration was measured as described by an in-house ELISA using a 12G2 antibody, kindly donated by Prof John Lewis[58].

### Glucocorticoid analysis
Total and free mouse serum corticosterone concentrations were determined by liquid chromatography–tandem mass spectrometry (LC-MS/MS) after modified equilibrium dialysis[59]. Briefly, the cap of a sterile 0.5 mL polypropylene Eppendorf tube was filled with 100 µL of PBS, and PBS-equilibrated 3.5 K MWCO dialysis membrane (Thermo Fisher, #88244) was laid upon the PBS-filled lid. The base of the Eppendorf tube was cut off and it was placed upside down on the covered lid, thereby fixing the membrane. Undiluted mouse serum (100 µL) was added on top of the dialysis membrane in the cut-off Eppendorf tube, which was sealed with parafilm and incubated overnight, at 37 °C, to ensure equilibrium dialysis of undiluted serum was achieved at physiological temperature. Dialysed serum sample was then removed from the vial for total corticosterone measurement, and dialysates were harvested from tube lids for free corticosterone measurements.

Human serum (100 µL) and dialysed/dialysate mouse samples (100 µL) were prepared in single ($n = 1$ technical replicate) alongside calibration standards (covering a range of 0.025–500 ng/mL) in a 96-well plate enriched with internal standard (Human, 10 ng, epi-cortisol; mouse, 10 ng, 13C-corticosterone) and diluted with 0.1% formic acid in water (100 µL) on an Extrahera liquid-handling robot (Biotage, Uppsala, Sweden). Diluted samples were transferred to an SLE + 200 extraction plate (Biotage, Uppsala, Sweden) and eluted into a collection plate using dichloromethane/propan-2-ol (98:2; 4 × 450 µL). The eluate was dried and reconstituted in water/methanol (70:30; 100 µL) before injecting directly from the 96-well plate for LC-MS/MS analysis.

Adipose tissue samples (100–250 mg) were enriched with internal standard (Human, 0.5 ng epi-cortisol; mouse, 0.5 ng 13 C -corticosterone) and homogenised (TissueLyser II, Qiagen) in acetonitrile with 0.1 % formic acid (500 µL). Calibration standards were prepared alongside the samples covering a range of 0.0025–10 ng. The samples were centrifuged and the supernatant (500 µL) was transferred to an ISOLUTE PLD + 96-well plate cartridge (Biotage, Uppsala, Sweden), subjected to positive pressure, eluent collected, and dried under nitrogen gas (40 °C). The samples were reconstituted in $H_2O$: MeOH (70:30; 100 µL) and sealed before analysis.

Extracts were analysed by LC-MS/MS on a Shimadzu Nexera X2 connected to a QTrap 6500+ mass spectrometer (AB Sciex) adapted from earlier methods[60,61]. Standards and samples were injected (20 µL) onto a Kinetex C18 column (150 × 2.1 mm and 2.6 µm; Phenomenex, #TN-1063) fitted with a 0.5 µm Ultra KrudKatcher (Phenomenex, #00F-4783-AN) at a flow rate of 0.3 mL/minute. The mobile phase system comprised water with 0.05 mM ammonium fluoride and methanol with 0.05 mM ammonium fluoride. The mass spectrometer was operated in positive ion electrospray ionisation mode using multiple reaction monitoring of steroids and internal standards. Mass transitions, voltages and retention times are detailed in Supplementary Table 3. The instrumentation was operated using Analyst 1.6.3 (AB Sciex, Warrington) and quantitative analysis of the data was carried out by least squares regression of the peak area ratio of the steroid to the corresponding internal standard with equal or 1/x weighting using MultiQuant software v3.0.3 (AB Sciex, Warrington).

### CBG binding capacity
Serum CBG was measured using a radioligand saturation assay that measures CBG concentration based on corticosteroid-binding capacity[62]. For in vitro studies serum samples treated with a combination of NE and AAT were further diluted 1:100 in PBS supplemented with 0.1% w/v gelatin and stripped off endogenous steroids for 30 min at room temperature with dextran-coated charcoal (DCC). For direct determination in experimental serum from mice and humans, samples were diluted (1:500) in the same PBS-DCC slurry as above and incubated for 30 min at room temperature, followed by centrifugation to sediment the DCC. Mouse supernatants were then incubated with 2nmol/L 1, 2, 6, 7 [3H4]-corticosterone in the absence or presence of excess unlabelled corticosterone (Steraloids; catalogue #Q1550-000) for total and nonspecific binding, respectively. Human supernatants were incubated with 2nmol/L 1, 2, 6, 7 [$^3H_4$]-cortisol in the absence or presence of excess unlabelled cortisol (Steraloids; catalogue #Q3880-000) for total and non-specific binding, respectively. After separation of free [3H]-steroid by adsorption with a DCC slurry for 10 minutes and centrifugation at 0°C, the amount of CBG-bound [3H]-steroid in the supernatant was determined in a scintillation counter, representing the corticosteroid-binding capacity of CBG (expressed in nmol/L). Inter- and intra-assay variability were both < 10%.

### Extraction and quantification of mRNA by RT-qPCR
Total RNA was extracted from adipose and liver using a RNeasy Mini kit (Qiagen Inc., Valencia, CA, USA) according to the manufacturer's instructions. The tissue was mechanically disrupted in either QIAzol

(Qiagen) for adipose tissue or RLT buffer (Qiagen) for liver tissue. cDNA was synthesised using a QuantiTect Reverse Transcription kit (Qiagen) according to the manufacturer's instructions. A quantitative real-time polymerase chain reaction was performed using a Light-Cycler 480 (Roche Applied Science, Indianapolis, IN, USA). Primers were designed using sequences from the National Centre of Biotechnological Information. The qPCR primer sequences are included in Supplementary Table 4. Samples were analysed in triplicate and amplification curves plotted ($y$ axis, fluorescence; $x$ axis, cycle number). Triplicates were deemed acceptable if the standard deviation of the crossing point was < 0.5 cycles. All primers were previously calibrated and used at efficiencies between 90–110%. Data analysis was performed using the Pfaffl method[63], with the abundance of each gene expressed relative to the mean of two housekeeping genes (Mouse, *Hprt, 18S;* Human, *PPIA, 18S).*

### Statistical analysis
All data presented are mean ± SD. Analyses were performed using Prism 8 software (GraphPad). Multiple groups were compared using Two-way ANOVA, repeated measures (RM) Two-way ANOVA, or mixed-effects model as appropriate. Multiple comparison post-hoc tests were carried if a significant effect was observed from AVOVA/mixed-effect model. Comparisons between two groups were assessed by paired or unpaired Student's t test as appropriate. $P$ value ≤ 0.05 were considered as statistically significant.

### Reporting summary
Further information on research design is available in the Nature Portfolio Reporting Summary linked to this article.

## Data availability
Data supporting the findings of this study are available within the paper and its Supplementary Material. Additional data supporting the findings described in this manuscript is available from the corresponding author upon request. Source data are provided with this paper, and can also be accessed at https://doi.org/10.7488/ds/7857 Source data are provided with this paper.

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

## Acknowledgements

Funding for this project includes a British Heart Foundation Intermediate Basic Science Research Fellowship to MN (FS/18/20/33449) and a Wellcome Senior Investigator award to BRW (107049/Z/15/Z). LC-MS/MS data were obtained at the University of Edinburgh, Edinburgh Clinical Research Facility, Mass Spectrometry Core, RRID:SCR_021833 with the AB SCIEX QTRAP 6500+ system, RRID:SCR_021831. We thank Tricia Lee, Jo Simpson and Scott Denham of the Mass Spectrometry Core for their technical assistance and acknowledge the financial support of NHS Research Scotland (NRS) for the Mass Spectrometry Core, Edinburgh Clinical Research Facility. We thank the Edinburgh Metabolic Phenotyping Facility, University of Edinburgh, for the skilled help and support.

## Author contributions

L.D.B., R.H.S., B.R.W. and M.N. conceived and designed the studies. L.D.B. performed clinical study. L.D.B., M.M., N.Z.M.H. and R.A. analysed clinical data. A.M., M.P., E.V., L.I. and M.N. performed mouse studies. A.M., M.P, J.N.C.T., B.N. and M.N. performed experiments and analysed data. G.L.H., R.H.S., B.R.W., M.N. interpreted results. M.N. prepared figures and draughted the manuscript; V.V., G.L.H., R.H.S., B.R.W. and M.N. edited and revised the manuscript.

## Competing interests

The authors declare no competing interests.
