## [Peer Review File · Nature Communications]

The NE/AAT/CBG axis regulates adipose tissue glucocorticoid exposure.

Corresponding Author: Dr Mark Nixon

Version 0:

Reviewer comments:

Reviewer #1

(Remarks to the Author)

Several previous studies suggested that neutrophil elastase (NE) and its inhibitor alpha1-antitrypsin play an important role in development of obesity-associated metabolic dysregulation but the underlying mechanisms remain incompletely understood. To change that, Boyle et al performed a combination of murine and human studies and suggest that this effect is at least partially exerted via the NE-mediated cleavage of corticosteroid-binding globulin. Unfortunately, the story has several important limitations that decrease its impact:

- The observed effects are seen only in one mouse model and only in male sex. The authors should try to confirm the findings in an independent model.
- The basic observations (i.e. less metabolic dysregulation in ELA-KOs) have been reported previously and because of that, the findings are of limited novelty. In addition, the usage of adrenalectomized animals seems to be an imperfect model since the procedure constitutes a major disruption of the endocrine signaling. Supplementation of antitrypsin or use of neutrophil elastase inhibitor that can be added at different stages of the experiment would be more convincing proofs.
- While the combination of mouse and human data is interesting, the assessed groups seem to be somewhat different. In particular, the subjects with heterozygous AAT mutation tend to have higher BMI values that might be in part responsible for some of the observed findings.
- It would be useful to include serum CBG levels in the assessed humans in table 1.
- To further corroborate the murine findings, the human data should be increased to allow a statistically meaningful assessment of both sexes.
- The available data (such as the large dataset from UK Biobank) suggests that subjects with a heterozygous AAT mutation are neither obese nor are they predisposed to metabolic syndrome. How do the authors explain this discrepancy?
- The "AAT group" assessed in the study seems to be genetically heterogeneous and does not contain only subjects with a heterozygous but also with a homozygous mutation (i.e. Pi*SS). Therefore, the labelling of the group should be adjusted to reflect this fact. In addition, the exact composition of the "AAT group" should be disclosed and its heterogeneity mentioned as an important caveat.

Reviewer #2

(Remarks to the Author)

In this manuscript Boyle et al provide data on the role of NE and AAT to modulate tissue glucocorticoid exposure in mice and humans.

The topic is interesting and important as hypercortisolemia is causally associated with cardiovascular disease and the NE/AAT/CBG axis may participate in the link between inflammation and adverse metabolism.

The analysis of the NE/AAT/CBG axis in metabolism, both in mice and human, is original although the role of NE/AAT in metabolism was reported before as well as the role of CBG in obesity. Thus, this study makes a link between these 2 sets of literature.

There are a number of concerns in the methodology, the data provided and their interpretation as detailed below which have led to overstatements of the results. Some parts of the manuscript are unclear due to either lack of details in the methodology

(Figure 3), the presentation of data split in the text but together on the figure (Figure 5) and mistake in the figure legend (Figure 6). The cited publications do not cover all the available literature.

- 1) Introduction line 36: "physiological" stress is odd. Glucocorticoids respond to very different types of stress, eg psychological stress trigger the highest activation of the HPA .
- 2) Introduction line 61 : this ref of Minni ,2012 (PMID: 22930537) is a good example of active role of CBG to deliver glucocorticoids to target tissue.
- 3) Introduction line 66: cleavage of CBG by NE was first shown by Pemberton 1988: PMID: 3143075
- 4) Figure 1& method lines 411-426: As the authors write themselves NE influences CBG affinity for corticosteroids. Yet, they analyzed CBG maximal capacity but not affinity. Indeed, a single concentration of corticosterone (20nM) was used while if increasing concentrations were tested and a Scatchard analysis performed, CBG affinity constant could have been measured. Given the high intra and inter-variability of the CBG maximal capacity assay (not provided here in the method section), such analysis is recommended.
- 5) Figure 1 and discussion lines 370-373: the impact of NE cleavage of CBG from pooled sera could be proven by running a SDS-PAGE showing CBG fragments as described in Sumer-Bayraktar 2016 Figure 2A (PMID: 27339896)
- 6) Figure 1e, 1f and method: free corticosterone fraction in vehicle was found to be around 50-60%. One would expect this fraction to be around 5% as CBG binds 85% of corticosteroids and albumin binds part of the 15% non-CBG bound. Indeed, in Figure 3c the WT CBG capacity is around 3%. Unless these 50-60% of free cort fraction is explained by the authors, these results of NE and AAT on cort free fractin are not valid.
- 7) Figure 1, 3: in serum corticosterone is distributed as CBG-bound, albumin-bound and free. It would be interesting to estimate albumin-bound corticosterone as described by Hammond, 1982,PMID: 6215538.
- 8) Figure 1: why is NE used at 4U for 1a-c and then 2U for 1d-f? it would be more homogenous to use the same quantity of NE throughout.
- 9) Supp Fig 1: CBG Bmax decreases between NE 2U and 4U but free cortisol fraction does not, please explain.
- 10) Lines 105-111 & Supp Fig2: Why is NE measured in serum and fat tissue in males and not in females after HFD exposure? Even if females ELA-KO do not show metabolic alterations it would be a good control to show that NE alterations found in males under HFD are not present in HFD fed females.
- 11) Figure 3 and Supp Figure 3: Data on serum corticosterone levels are provided in both figure 3a and Suppl Fig3c for male and this is confusing. In Suppl Fig 3c one understands that the measures were done at 7 am and 7pm after 7 weeks of diet. In Figure 3a there is no description about the conditions of sampling and the data are very variable from 50nM to 170nM in WT males, i.e. in between the data of am and pm provided in Suppl 3c. Thus, data from Fig 3a may reflect circadian variations from a non-time-controlled blood sampling. The animals with corticosterone concentrations above 150nM in figure 3a look like stress concentrations, the same in Suppl Figure 3c concentrations are expected to be less than 50 nM for am and less than 150 nM for pm in unstressed mice (see eg Gray 2015 PMID: 25559007). The conditions of mice housing conditions, euthanasia and sampling should be described in the Method section: time of light on/off, time of the day for blood sampling in each experiment, time between handling the animal and blood collection (if > to 3 min = stress levels), conditions of euthanasia= use of analgesics/anesthetics or not (this greatly influences cort levels).
- 12) Lines 150-151 fig 3f-3g: $p=0.07$ so adipocytes tend to be reduced in size in male ELA-KO vs WT. PPAR γ gene expression may be interesting to measure in the groups.
- 13) Lines 145-147: "in males" should be added in the sentence as females were not tested for these measures
- 14) Figure 3: how are CBG Bmax and affinity within adipose tissue?
- 15) Line 154 and Fig3i: Why are Per1 levels so variable in WT? this is unusual and not seen in liver of the same animal. Fkbp5, Lpl , Atgl & Adipoq are also GC responsive genes but they show no differences between groups, this should be mention in the text. Ccl2 is not significantly different between groups: to be removed in the text as "significant alterations in pro-inflammatory markers"
- 16) line 173: "having demonstrated an association": please rephrase as one cannot "demonstrate" an association. Furthermore, the association between NE-deficiency and reduced glucocorticoid action is an overstatement at this stage.
- 17) Line 173-189: adrenalectomy is a very unspecific way to assess the importance of glucocorticoids in the metabolic protection of ELA-KO male mice. Given the question under study (NE/AAT/CBG axis), one would expect an experiment were CBG would be specifically modulated in vivo by a specific knock-down (eg through shRNA) or the use of CBG KO.

18) Figure 4: the statistics are not provided for graphs 4a, 4c. For the other graphs of Figure 4, interaction and main effects of the 2 way ANOVA are not provided. Interaction has to be significant before running post-hoc tests. Are ELA-KO sham and ELA-KO ADX significantly different for weight gain? AUC Glucose or HOMA-IR? If not the removal of adrenals (and glucocorticoid levels even less) does not play a role for these parameters.

19) Lines 183-184: this assertion is not convincing: it is not significant in sham ELA-KO while it is significant in all other groups (4e); HOMA-IR does not seem different between sham and ADX ELA-KO.

20) Line 193: "having established.." is an overstatement.

21) Line 198: Isn't there a word missing in the sentence?

22) For the clinical study, one would expect that males would be preferentially recruited given the animal data, however 75% of AAT patients were females. Is there a sex difference in the prevalence of the disease or the recruitment was done before the animal data were available? Although BMI and fasting insulin are not different between controls and AAT patients statistically according to Table 1, AAT patients have on average a BMI of obese and show hyperinsulinism. Insulin down-regulates CBG as shown by Fernandez-Real (PMID: 12364459). Is there a correlation between insulin and cortisol levels or BMI and cortisol levels in the subjects, especially the AAT patients? Il-6 is another factor that regulates CBG and might be elevated in AAT patients given their BMI. Its influence on CBG and cortisol needs to be tested.

23) Line 206 onwards: from here, it is very difficult to jump from the text to figure 5 as in the text the placebo arm only is considered but on the figures all groups (placebo and CRASH treated patients) are shown. The CRASH data are presented in the text much later, after figure 6, this is very confusing. Again, as in Figure 4, no stats are provided for graphs 5a, 5b and 5c so one cannot see whether "circulating free fraction is significantly elevated in AAT-deficient heterozygotes" (line 208). Details of interaction and main effects, then post hoc tests should be provided.

24) Figure 5e: the data for LPL are not very convincing as 3 AAT patients only (over 16) have high levels. Similar to Figure 3i, glucocorticoid responsive genes such as ATGL or ADIPOQ are glucocorticoid targets but show no differences between groups. One or 2 AAT patients seem to have elevated levels of all transcripts except Adipoq: could this be explained by sex (as there are only few males in the patient cohort) or by AAT levels (a correlation could be done between AAT levels and levels of the various transcripts).

25) Figure 6: the figure legend does not fit with the figure and the text, which is very confusing. It might be useful to add "plasma" and skeletal muscle" above the corresponding graphs in addition to correct the legend. Plasma: Two AAT patients have high free cortisol levels: are they males? How are their insulin levels? free cortisol fraction is not shown, is it because the difference between groups is not significant? Skeletal muscle: this time free cortisol fraction is shown (and convincingly different between groups)) but what about free cortisol levels? how are CBG B max levels in this tissue?

26) Line 304: no correlation analyses were done in the study

27) Line 322: no data in adipose tissue are presented in the manuscript.

28) In the method, serum CBG binding capacity is described twice: (lines 411-426 and lines 601-613) each time mentioning mouse and huma assays. However, in the 2nd description supernatants were incubated with 2 nM of cortisol while it was 20 nM in the first description for human serum. For the statistics, the post hoc tests are not specified.

In conclusion, additional evidence and further details are needed to support the claims presented in the discussion and abstract. Indeed, the data of free cort fraction of figure 1 are unexplained, so the data are not valid as such. Figure 3 provides interesting data (graph 3h in particular) but the gene expression data are not fully supporting a higher glucocorticoid action. The adrenalectomy experiment is unspecific, other more specific approaches are expected and statistical details are missing to well interpret the data. In the clinical study, females represent 75% of the AAT heterozygote patients when the animal data show no effect of the NE deficiency in females. Thus the study is underpowered for males. Furthermore, the measure of glucocorticoids within adipose tissue was unsuccessful for technical reasons and therefore done on skeletal muscle but this tissue was not analyzed in mice. CBG Bmax and affinity were not measured within adipose or skeletal muscle and the potential role of local within tissue) CBG not even discussed regarding the work of M Grasa's group (eg Gulfo 2016, PMID: 27323695, reviewed in PMID: 3143075). CBG in the context of obesity is not discussed neither, regarding the work of JM Fernandez-Real (eg PMID: 12364459, PMID: 11586502, PMID: 10651759, PMID: 10487686). The AAT patients have a BMI of obese subjects on average (BMI=30) and high insulin levels (and maybe high Il-6 levels) so further analyses are required to dissect the role of each of these confounding factors.

Reviewer #3

(Remarks to the Author)

Boyle and colleagues provide an interesting study linking neutrophil elastase and its inhibitor alpha1 antitrypsin (AAT) with tissue glucocorticoid levels in a model of diet-induced obesity in mice. The study is very well performed and the results are clearly described. It is very positive to see efforts to test their hypothesis in both male and female mice; in addition, analysis of human samples in patients heterozygous for AAT and arterio-venous sampling in humans provides confidence for the

relevance of the proposed mechanism in humans. Despite my overall support for this study, I have some rather minor concerns and comments:

Major:

In figure 1 the authors should block elastase activity (in NE treatment) in panels a/b and assess the effect on binding capacity.

Neutrophils are the dominant source of Elane, yet I cannot find details on circulating and tissue counts of neutrophils in any of their models. These should be provided throughout the study.

In line with the above comment, neutrophil counts (and the release of Elane) is circadianly regulated. Do tissue corticosterone change over the day and are these in phase with local neutrophil counts?

In figure 3 levels of plasma lipids (triglycerides, cholesterol) should be provided.

For Elane concentration rather than activity is given in many panels (e.g. Suppl Fig 2a). Please also provide activity.

It is a striking finding that the authors' key finding does not seem to hold true in female mice. While this is very interesting, it is disappointing that there is no satisfying explanation. Would the Elane/CBG axis be operational in gonadectomized male mice?

Minor:

The correct abbreviation for neutrophil elastase gene in mice is Elane, not ELA.

Figures panels should be referred to in the correct order. The current order in some of the figures is rather confusing.

Version 1:

Reviewer comments:

Reviewer #1

(Remarks to the Author)

The authors should be commended for the extensive amount of work that they did. In the revised version, they were able to clarify a couple of inaccuracies and as a result, the data are clearer now. Despite that, some of the key findings remain somewhat puzzling, i.e. (i) the fact that the difference is seen in males only and cannot be reproduced in an independent model; (ii) the fact that AATD subjects seem to be neither obese nor suffer a metabolic syndrome. These discrepancies need to be clearly disclosed and the results from the NE inhibitor experiments should be mentioned in the manuscript.

Reviewer #2

(Remarks to the Author)

In this revised manuscript, the authors have improved the form and they have answered part of the questions but not all.

-Figure 1d and 1e: the authors have left these data showing a percentage of free cort from 50% to 80% which I believe is not correct or understandable. They explain that this artefact come from the fact that the sera were diluted 1:50 prior to treatment with NE/AAT and determination of CBG Bmax and free steroid levels. They claim that this dilution is required because of "the necessary stoichiometry to facilitate NE-mediated cleavage of CBG" and they give as a reference PMID 2370299. In the latter publication, I do not find any mention of stoichiometry necessary for CBG cleavage by NE that would support the serum dilution. Diluting the serum sample will indeed disturb the equilibrium between bound and unbound steroids. Since the free fractions shown are an artefact, probably due to the serum dilution, I think these data cannot be presented as such and the same experiments should be redone on undiluted serum.

Figure 3a : The corticosterone levels found (mean around 100 nM?) cannot reflect basal morning levels but most probably cort levels of stressed mice. I believe mass spectrometry vs ELISA method cannot explain these very high levels as claimed by the authors. As a proof, these levels are much higher than those measured in Suppl Figure 3d for AM (WT:16.3nM and ELA-KO: 38.3 nM) by Mass Spec by the same authors. Therefore Figure 3a, 3b and 3c I believe are not correct.

-Figure 3j: Glucocorticoids being anti-inflammatory isn't it strange that the levels of pro-inflammatory markers (Tnf α , Cd68, Itgax) are down-regulated ? could it be discussed in the discussion together with the surprising up-regulation of Pparg while adipocyte size tend to be decreased?

-The authors claim that CBG Bmax cannot be measured within tissue. However, there are publications in which CBG Bmax was measured within adipose tissue: PMID 11507677 and 11855738. In this study the evaluation of CBG Bmax within gonadal fat is essential.

-The response to the reviewers about the unspecific action of adrenalectomy regarding glucocorticoids involvement is not entirely satisfactory. The authors agree that manipulation of CBG would be a better option than adrenalectomy. They explain that they did not use this option because CBG-KO display very low levels of glucocorticoids. However, they did not discuss my suggestion of using an adipose-specific knock-down of CBG through virus-mediated expression of a shRNA against CBG at adulthood. In this model the HPA axis would not be disturbed as in the full CBG-KO because the manipulation will be localised within adipose tissue and at adulthood with no time for compensatory mechanisms.

-Figure 5: Free cort is increased but there is no change in CBG Bmax. The hypothesis is that CBG Bmax is decreased in adipose tissue. Again the measure of CBG Bmax within adipose tissue is necessary to the demonstration.

-Figure 6d: adipose cortisol is found increased in AAT patients under placebo but not after CRASH (Control CRASH vs AAT CRASH). When separated by sex (Supp Figure 13), the adipose cortisol seems to be up-regulated in male patients only, whether under placebo or after CRASH. In the rebuttal letter, the response Figure 4 shows that the increase in Per1 and LPL are driven by female patients. Together these data do not look congruent.

- The clinical study was not powered to analyse sex effects. Thus the regression analysis showing no significant effect of sex should not be reported.

-The authors claim that their data "underscore the potential therapeutic opportunities to manipulate glucocorticoids and reduce cardiometabolic risk" but in the rebuttal letter they answered to reviewer 1's remark that AAT patients are neither obese nor predisposed to metabolic syndrome, because of likely compensation for the altered glucocorticoid profile, eg due to reduced whole-body 11bHSD1 activity.

Although the study presents some data that are in favor of the authors' hypothesis that The NE/AAT/CBG axis regulates adipose tissue glucocorticoid exposure, essential experiments are missing for claiming "compelling evidence". Additionally, the quality and interpretation of some of the data do not entirely meet the standards of the literature.

Reviewer #3

(Remarks to the Author)

The authors have addressed most of my comments; yet, two questions have not been addressed in full:

1. Neutrophils are the dominant source of Elane, yet I cannot find details on circulating and tissue counts of neutrophils in any of their models. These should be provided throughout the study.
Here the authors argue that blood counts are not available; however, I was specifically asking for tissue counts and given that tissues stainings are shown in figure 3, I assume these sections should still be available.
2. The correct abbreviation for neutrophil elastase gene in mice is Elane, not ELA.
This is still not changed throughout, especially in the figures.

Version 2:

Reviewer comments:

Reviewer #1

(Remarks to the Author)

The study limitations that were the main outstanding issues, are now properly disclosed.

Reviewer #2

(Remarks to the Author)

In this revised version of the manuscript, the authors have provided new explanations to clarify the points raised in the previous reviewing. As elaborated before, there are interesting and novel data in this study but also puzzling and unconvincing ones. Overall, the conclusions of the authors remain overstated at this stage.

The authors have now well explained the aim of the measures presented in Fig. 1 and they recognize that presenting their results as % free cort was equivocal. It is indeed better to present these results as fold change from vehicle as they did. To be perfectly unambiguous, the authors should add in the legend of FigS1 that free cortisol (%) values presented do not reflect physiological levels because serum samples were diluted.
As the authors wrote in the main text, these data on NE/AAT/CBG axis controlling glucocorticoid action are in vitro evidence.

Concerning basal/morning corticosterone levels in Fig3a I still disagree with the authors and I still believed that the data reflect stress values even if the values are lower than those obtained by restrain stress, they just reflect a milder stress than restrain. Consequently, the data of Fig.3b and 3c are also mild stress data. In Suppl. Fig.3d, where basal values are clearly obtained, what are the free cort % and free cort concentrations?

The tentative explanation about reduced proinflammatory markers in Fig.3j is far-fetched. The data remains puzzling. As already discussed in previous reviewing the gene expression data, that are purposed to show increased or decreased glucocorticoid action, are not convincing. The usual glucocorticoid responsive genes examined in many studies are Per1, Fkbp5, and Sgk-1. Here, Per1 only is tested but highly variable results are presented in WT animals for gWAT tissue so it is not enough. Then pro-inflammatory genes and fat synthesis gene (Pparg) are opposite to expectations. Fkbp5 was measured in sWAT, why not in gWAT?

As for CBG Bmax measured within fat tissue, I think mice can be perfused after short isoflurane anesthesia (that do not disturb circulating cort levels) with 1 ml of PBS in the heart to wash out blood contamination to all organs. It would have been worth trying.

Concerning CBG inhibition within fat tissue, I now understand that the hypothesis of the authors is that CBG in fat tissues derives from plasma rather than being synthesized in situ. First, this remains to be proven even if the authors do not believe previous published data.

Then, I understand that the general hypothesis (in mice) is that although serum-derived AAT is present within adipose tissue the increased NE levels, caused by neutrophil infiltration, overcome AAT inhibition. This is not demonstrated here. Could AAT be measured within adipose tissue? If NE/AAT balance is really increased in adipose tissue compared to serum, could AAT be added specifically within adipose tissue of HFD-fed male mice to rescue the phenotype? Another possibility would be to add the NE inhibitor (GW311616A) locally in adipose tissue of HFD-fed male mice in order to prove the local action of NE in gWAT and increased glucocorticoid exposure.

Such NE increase in gWAT of male mice fed HFD is not observed in females fed the same diet. However, I wonder if this does not reflect a floor effect since females show much reduced (if any?) metabolic dysregulation after 8 weeks of HFD consumption. Since there is no comparison with females fed a standard chow, it is difficult to conclude. A longer exposure to HFD might reveal similar NE increased in females too.

It is still puzzling that in AAT^{+/-} patients, free cort % is increased in the circulation but not corrected by the HPA axis feedback mechanism. Furthermore, CBG levels are unchanged but they predict free cort levels by regression analysis. How can this be explained?

In Fig.6d the increased adipose cortisol is driven by 4 patients out of 16. For Per1 and LPL transcripts the increased seems also driven by few patients. Are they the same patients for cort and transcripts high values? Could a correlation be calculated? If significant this would mean that another factor than AAT influence cortisol levels in AAT^{+/-} patients.

The abstract does not reflect the data obtained in the study:

- 1) it is odd to read the sentence "Here we used complementary approaches in mice and humans to show that male mice lacking NE fed a high-fat diet have reduced glucocorticoid exposure in adipose tissue, with improved glucose tolerance and insulin sensitivity". The most important findings are not presented and the females results ignored.
- 2) "The protective effect of NE deficiency is lost when endogenous glucocorticoids are removed" is not correct since whole adrenals are removed. Furthermore, this result is not a major point in the story.
- 3) Yes AAT^{+/-} patients seem to have higher circulating free cort but the latter is unexplained given than neither CBG nor feedback control are impacted.
- 4) Yes adipose and skeletal glucocorticoid exposure look increased but maybe only in part of patients (4/16). The mechanism of this increased exposure is different from mice as the free cort fraction is increased systemically.
- 5) "These data demonstrate that NE and AAT modulate local tissue glucocorticoid bioavailability in vivo, revealing a novel mechanism linking inflammation and metabolism. » To me, as discussed above, there is no data to claim such "demonstration" of "local tissue glucocorticoid bioavailability regulation by NE and AAT".

In my opinion, what can be claimed at this stage is that adipose cort levels are increased in gWAT of HFD-fed male mice compared to standard chow fed animals and associated with increased NE specifically in this tissue. Additionally, NE total ablation (Elane^{-/-} mice) results in reduced cort levels in gWAT specifically. Together these 2 results are indeed very interesting. It remains to demonstrate that 1) NE/AAT levels in gWAT are sufficiently increased so that AAT no longer inhibit NE action on CBG and that cleavage of CBG within adipose (and maybe muscle) is increased too.

In humans the increased cort release in adipose and muscle of (some) AAT^{-/+} patients are very interesting but as discussed above further data are necessary to be convincing.

Reviewer #3

(Remarks to the Author)

The authors have addressed my concerns in full. I have no additional questions.

Version 3:

Reviewer comments:

Reviewer #2

(Remarks to the Author)

The additional data on glucocorticoid-responsive genes expression in both mice and humans is strengthening the hypothesis put forward. The measure of AAT in mice is also new data in favor of the hypothesis.

While I still believe that measurement of CBG binding activity in tissue without blood contamination is feasible, this important limitation is now presented in the discussion.

Similarly, AAT supplementation as done in the article D'Souza, R. F. et al. Am J Physiol Endocrinol Metab 321, E560-E570, seems feasible to me and would have provided further functional evidence of AAT/NE involvement. Prolastin-C (human α 1-antitrypsin A was provided free of charge by Grifols USA, Los Angeles, CA in D'Souza 's article.

The section on adrenalectomy still needs improvement in the writing because it still read as if adrenal removal was a proof of glucocorticoid involvement:

Eg line 185-186: "we tested whether the metabolic protection of *Elane*^{-/-} male mice was dependent on glucocorticoids » To me, the authors tested if it was dependent on adrenal presence.

Line 192-193 « Assessment of glucose tolerance revealed a similar outcome, with *Elane*^{-/-} mice lacking endogenous glucocorticoids having no further improvement in their glucose tolerance compared to WT littermates lacking glucocorticoids (Figure 4c and 4d) ».

Line 206 « Our murine data demonstrated that NE deficiency attenuates adipose glucocorticoid action » : to me there is no demonstration, I would suggest to write : Our murine data showed that NE deficiency is associated with reduced adipose glucocorticoid action »

In the discussion, from line 322 it should be mentioned more clearly that the data were found in male mice because as it is, it reads as if it was a general finding.

There are a number of typo errors, eg line 144, line 1440

Authors' responses

Manuscript ID: NCOMMS-23-43150-T

'The NE/AAT/CBG axis regulates adipose tissue glucocorticoid exposure'

Reviewer 1:

'The observed effects are seen only in one mouse model and only in male sex. The authors should try to confirm the findings in an independent model. Supplementation of antitrypsin or use of neutrophil elastase inhibitor that can be added at different stages of the experiment would be more convincing proofs.'

We thank the reviewer for their suggestion to extend our murine work to further test our hypothesis. Having observed a protective effect on metabolic health and reduced adipose glucocorticoids in male ELA-KO mice, we sought to determine if the same protective phenotype, and change in tissue glucocorticoids was present in female ELA-KO mice. Intriguingly this was not the case. We initially postulated that this may have been due to the established 'metabolic protection' in female vs male mice observed in the literature. In our revised manuscript, we provide additional data demonstrating that female mice do not upregulate NE in their visceral adipose tissue (Supplementary Figure 2), suggesting a potential underlying mechanism for this sexually dimorphic metabolic protection. Together with our *in vitro* data and our human model of AAT-mediated NE manipulation, we believe this provides complimentary evidence that NE modulates glucocorticoid action across mouse and human.

However, we take the reviewer's point, and to attempt to answer this, we have tried two independent approaches in line with the reviewers suggestion of using a commercially available and previously published neutrophil elastase inhibitor (GW311616A) (Response Figures 1 and 2 below). In both instances, we administered 2mg/kg GW311616A by oral gavage, a dose previously shown to effectively inhibit NE activity in mice (PMID: 23562077, 32278493). Our initial approach aimed to determine the effect of NE inhibition on glucose and insulin dysregulation established by 8 weeks of HFD (Response Figure 1). Here we fed adult male C57Bl6/J mice a HFD (58% kcal fat with sucrose) for 8 weeks, before commencing co-administration of GW311616A every other day for an additional 4 weeks. (Response Figure 1a). However, despite following the same protocol and sampling regime (PMID: 23562077), we found that serum NE activity was not suppressed (Response Figure 1b), and improvements in diet-induced body weight gain (Response Figure 1c) or glucose tolerance (Response Figure 1d) were not observed. To determine if this was a consequence of an already established presence of NE and metabolic dysregulation resulting from chronic dietary manipulation, we have since investigated an alternative study design in which we co-administered GW311616A alongside HFD from the beginning of the dietary intervention (Response Figure 2a). Surprisingly, we found a similar result as before, with no inhibitory effect of GW311616A on serum NE activity (Response Figure 2b), or on measures of body weight gain (Response Figure 2d) or glucose tolerance (Response Figure 2e). While we accept that these results do not add to the proof that NE action in diet-induced murine obesity regulates adipose corticosterone exposure, we do not believe this diminishes the validity of the current data presented within the manuscript, and that our extended experimental approaches *in vitro* and *in vivo*, combined with additional analyses in both murine and human paradigms of genetic NE manipulation, provide the necessary further validation of proof. We have not included the data below within the revised manuscript, but are more than happy to should the reviewer and/or editor feel it would be valuable to the readership.

Response Figure 1. **a**, study design indicating adult (8 week old) male C57Bl6/J mice were fed a high-fat diet (58% kcal fat with sucrose) for 12 weeks. After 8 weeks of dietary manipulation, mice were co-administered the NE inhibitor (GW311616A; 2mg/kg) or vehicle (H₂O) every other day by oral gavage. **b**, serum NE activity at cull (12 weeks) was unchanged between mice receiving NE inhibitor (purple diamond) or Vehicle (green circle). **c**, body weights measured weekly over the course of treatment showed no change, and **d**, glucose tolerance tests performed between week 11 and 12 showed no change between groups. N=8 per group. Data were analysed by unpaired t-test or RM Two-way ANOVA.

Response Figure 2. **a**, study design indicating adult (8 week old) male C57Bl6/J mice were fed a high-fat diet (58% kcal fat with sucrose) and co-administered the NE inhibitor (GW311616A; 2mg/kg) or vehicle (H₂O) every other day by oral gavage for 8 weeks. **b**, serum NE activity at cull (8 weeks) was unchanged between mice receiving NE inhibitor (purple diamond) or Vehicle (green circle). **c**, corticosterone levels in gWAT were unchanged between groups. **d**, body weights measured weekly over the course of treatment showed no change, and **e**, glucose tolerance tests performed between week 7 and 8 also showed no change between groups. N=10 per group. Data were analysed by unpaired t-test or RM Two-way ANOVA.

The basic observations (i.e. less metabolic dysregulation in ELA-KOs) have been reported previously and because of that, the findings are of limited novelty. In addition, the usage of adrenalectomized animals seems to be an imperfect model since the procedure constitutes a major disruption of the endocrine signaling.

While the basic metabolic observations have been shown, the underlying mechanisms linking NE and diet-induced metabolic dysregulation have not been elucidated. Moreover, we provide the first *in vivo* evidence in both mice and humans of a potentially crucial mechanistic link between inflammation, glucocorticoids, and metabolism. Our use of adrenalectomy aimed to determine if the improved insulin sensitivity observed in male ELA-mice was glucocorticoid-dependent, supporting a role for adipose glucocorticoids in mediating the metabolic dysregulation. Manipulation of glucocorticoids presents a unique challenge given the multiple feedback mechanisms in place, particularly the HPA axis, that aim to correct any subtle manipulations. We and others have previously utilised adrenalectomy as a suitable approach to determine the importance of glucocorticoids in metabolic and neuropathological conditions in the absence of these confounding systemic or tissue 'corrections' (PMID: 27535620, 30292559). However, we are aware of the limitations of this approach and have thus tempered our interpretations to state that the phenotypes observed are adrenal- rather than glucocorticoid-dependent.

'While the combination of mouse and human data is interesting, the assessed groups seem to be somewhat different. In particular, the subjects with heterozygous AAT mutation tend to have higher BMI values that might be in part responsible for some of the observed findings.'

We apologise for not providing sufficient information on the assessed groups within the human study. As part of the new Supplementary material, we now include individual graphical representations for the data outlined in our revised Table 1 (Supplementary Figure 5). This includes the data referenced by the reviewer relating to BMI (specifically highlighted in the individual graph below, Response Figure 3). We appreciate that viewing the figures in tabular form may suggest that the AAT heterozygous group had higher BMI values, however as is clear from the graphical representation, this is somewhat skewed by a single individual within that group. We have robustly tested the statistical significance of our comparisons, and no differences between the groups are observed for BMI (or indeed any parameter with the exception of AAT and NE). Moreover, in our revised manuscript we have also carried out regression analyses to determine if BMI, sex, or fasting insulin acted as a significant predictor variable for free cortisol. None of these reached statistical significance (BMI $P=0.8171$, Sex $P=0.3239$, Fasting insulin $P=0.5542$). Interestingly the regression analyses found that AAT ($P=0.0378$) and CBG Bmax ($P=0.0028$) significantly predicted free cortisol. We have now added this data in tabular format as Supplementary Table 1, and added these analyses into our results section (Ln 255).

Response Figure 3. BMI comparison in subjects with heterozygous mutations in SERPINA1 (AAT+/-) and matched controls (Control). N=16 per group. Data were analysed by unpaired t-test (P=0.21).

'It would be useful to include serum CBG levels in the assessed humans in table 1.'

We agree and have now added this data to the revised Table 1 and included graphical representations for further clarity (Supplementary Figure 5).

'To further corroborate the murine findings, the human data should be increased to allow a statistically meaningful assessment of both sexes.'

We agree this would be helpful, unfortunately it is not feasible to expand our clinical study. However, we have undertaken additional analyses of the human data in order to examine any potential indications of sexual dimorphism in the responses. We performed regression analyses to determine if Sex predicts free cortisol in our study, and did not find any significant effect ($P = 0.3239$). We have included the results of these analyses in our new Supplementary Table 1. Moreover, in our revised manuscript we have incorporated sex-specific presentation of the baseline characteristics of the subjects (Supplementary Figure 7), the arterio-venous tracer study (Supplementary Figure 10), and the CRASH study (Supplementary Figure 13).

'The available data (such as the large dataset from UK Biobank) suggests that subjects with a heterozygous AAT mutation are neither obese nor are they predisposed to metabolic syndrome. How do the authors explain this discrepancy?'

The reviewer is correct that the SNPs relevant to our AAT heterozygous subjects ('Z' rs28929474; 'S' rs17580) do not associate with readouts of obesity or overt metabolic syndrome. In the case of rs28929474, there are associations with increased trunk mass and trunk fat mass, however the predicted effect size is small. In our study, the groups (Control and AAT heterozygotes) were well matched for measures such as BMI and fasting glucose/insulin. Together with the UK BioBank data, this suggests limited influence on obesity and other cardiometabolic measures. In our manuscript, we have therefore steered clear of any overt suggestions that subjects with these mutations are predisposed to metabolic syndrome. However it is important to consider that there is likely compensation for the altered glucocorticoid profile we observe in these subjects. For example our tracer study revealed reduced whole-body 11β HSD1 activity, indicating reduced

extra-adrenal cortisol generation (most likely in the liver which is a key metabolic tissue). This compensatory mechanism would potentially limit the adverse cardiometabolic effects caused by increased adipose glucocorticoid exposure, making associations at a population-wide level difficult to determine.

'The "AAT group" assessed in the study seems to be genetically heterogeneous and does not contain only subjects with a heterozygous but also with a homozygous mutation (i.e. Pi*SS). Therefore, the labelling of the group should be adjusted to reflect this fact. In addition, the exact composition of the "AAT group" should be disclosed and its heterogeneity mentioned as an important caveat.'

We apologise for this typographical mistake, and for the lack of clarity on the group composition. The AAT heterozygous group included either PiMS or PiMZ mutations only. No subjects with the homozygous mutation PiSS were recruited. We have amended the text (Ln 563) to indicate the number of each genotype within the AAT heterozygous group. Moreover, following this comment we performed additional analyses, separating our AAT heterozygous group by genotype. We now include graphical representation of these analyses in our new Supplementary material, including genotype-specific presentation of the baseline characteristics of the subjects (Supplementary Figure 8), the arterio-venous tracer study (Supplementary Figure 11), and the CRASH study (Supplementary Figure 14). We have also added a comment in the discussion reflecting the potential impact of the group heterogeneity on the results and their interpretation (Ln 370-379).

Reviewer 2:

Introduction line 36: "physiological" stress is odd. Glucocorticoids respond to very different types of stress, eg psychological stress trigger the highest activation of the HPA.

We have amended the text, replacing 'physiological' with 'stress and trauma' to better reflect the activation of the HPA axis.

Introduction line 61: this ref of Minni ,2012 (PMID: 22930537) is a good example of active role of CBG to deliver glucocorticoids to target tissue.

We agree that it is important to highlight the experimental evidence that CBG acts as a regulator of the systemic glucocorticoid 'reservoir', enabling the body to respond. We have now added in the suggested reference.

Introduction line 66: cleavage of CBG by NE was first shown by Pemberton 1988: PMID: 3143075

We have corrected the reference as indicated.

Figure 1& method lines 411-426: As the authors write themselves NE influences CBG affinity for corticosteroids. Yet, they analyzed CBG maximal capacity but not affinity. Indeed, a single concentration of corticosterone (20nM) was used while if increasing concentrations were tested and a Scatchard analysis performed, CBG affinity constant could have been measured. Given the high intra and inter-variability of the CBG maximal capacity assay (not provided here in the method section), such analysis is recommended.

We apologise for not providing sufficient detail in the methods and text for these points. Our inter- and intra-assay variability of the CBG binding capacity are both <10%, in line with the original method description (PMID: 6193907), and we have added this information to the methods (Ln 698). CBG maximal binding capacity and affinity are distinct but related parameters. Steroid-binding capacity measurements rely on the saturation of a single steroid-binding site per CBG molecule, and the amount of bound ligand is therefore assumed to directly reflect the molar concentration of the protein. In our steroid-binding capacity assay, DCC is used to adsorb non-CBG bound radiolabelled ligand, allowing for the identification of samples in which CBG has altered steroid-binding affinity. We have presented data regarding binding capacity (B_{max}) because from a technical viewpoint, determination of the very low affinity of CBG after cleavage by NE is almost impossible using the charcoal separation method because the dissociation of the steroid from CBG occurs too rapidly. However, in light of this we have added a line to our discussion to emphasise that our functional measurements of CBG (i.e. B_{max}) has inherent limitations (Ln 423-426).

Figure 1 and discussion lines 370-373: the impact of NE cleavage of CBG from pooled sera could be proven by running a SDS-PAGE showing CBG fragments as described in Sumer-Bayraktar 2016 Figure 2A (PMID: 27339896)

Relating to Figure 1, there is a well established literature on the impact of NE on CBG cleavage, with several publications demonstrating that treatment of affinity purified CBG from pooled sera with NE results in a downward shift of the CBG band as observed on a Western blot (PMID: 2370299, 24848868, 29273683, 21875666). Thus from an initial 'in vitro' perspective, we believe it is more important to demonstrate an NE-mediated suppression of CBG binding capacity and increase in free glucocorticoids. From an 'in vivo' perspective, we agree that alongside our serum CBG binding capacity data, a Western blot would provide additional evidence supporting/refuting a role for systemic NE manipulation of CBG. However, as is outlined in the manuscript referenced by the reviewer, accurate assessment by this method requires affinity purification of CBG from a volume of pooled sera (in the case of human samples this was previously obtained from a commercially available source, Affiland, that is no longer available). Unfortunately paucity of sample prevents us from doing this for either our murine or clinical samples. In light of this, we have amended the text in our discussion to better highlight the limitations of our approach and indicate that future studies would benefit from concomitant analyses of multiple CBG readouts where possible (Ln 416-437).

Figure 1e, 1f and method: free corticosterone fraction in vehicle was found to be around 50-60%. One would expect this fraction to be around 5% as CBG binds 85% of corticosteroids and albumin binds part of the 15% non-CBG bound. Indeed, in Figure 3c the WT CBG capacity is around 3%. Unless these 50-60% of free cort fraction is explained by the authors, these results of NE and AAT on cort free fraction are not valid.

We apologise for not providing sufficient detail relating to the measurement and interpretation of these results, and have added in this additional explanation to the Methods (Ln 467-470). In our initial 'in vitro' experiments (Main text Figure 1), serum samples were diluted (1:50) prior to treatment with NE/AAT and determination of CBG binding capacity or free steroid levels. This dilution is required because of the necessary stoichiometry to facilitate NE-mediated cleavage of CBG (PMID: 2370299). In undiluted conditions, the concentrations of NE needed to cleave CBG in serum are not achievable through addition of commercially available NE. By diluting the serum, one can still determine the ability of regulatory mechanisms (such as proteolytic cleavage) to alter CBG binding capacity in biological samples. However, dilution of serum does have an inherent limitation in that it disturbs the equilibrium of bound and unbound steroid, establishing a new set-point. The result is that untreated, baseline serum samples present with significantly elevated percentage free fractions (approximately 50% free glucocorticoid).

Figure 1: why is NE used at 4U for 1a-c and then 2U for 1d-f? it would be more homogenous to use the same quantity of NE throughout.

We thank the reviewer for picking up on this typographical mistake – NE was used at 4U for the experiments outlined in 1d-f. The text in the Methods (Ln 509) and the text in the figure legend (Ln 951-962), has been amended accordingly.

Supp Fig 1: CBG Bmax decreases between NE 2U and 4U but free cortisol fraction does not, please explain.

This is an excellent observation. The most likely explanation is that this may be due in part to the methodology behind free cortisol assessment (as indicated two points above). We believe that the serum dilution required in order to assess the effect of NE on free cortisol, and the subsequent higher 'baseline' of percent free cortisol, means that pushing the system beyond approximately 80-85% free cortisol is not possible. Therefore, the percent free appears to plateau, despite NE having a greater impact on CBG binding capacity.

Lines 105-111 & Supp Fig2: Why is NE measured in serum and fat tissue in males and not in females after HFD exposure? Even if females ELA-KO do not show metabolic alterations it would be a good control to show that NE alterations found in males under HFD are not present in HFD fed females.

We thank the reviewer for this excellent suggestion. We have performed the requested experiment in female mice, and now include the data in the revised Supplementary Figure 2 (panels e-g). This new data is also highlighted in the main text (Ln 111-119).

Figure 3 and Supp Figure 3: Data on serum corticosterone levels are provided in both figure 3a and Suppl Fig3c for male and this is confusing. In Suppl Fig 3c one understands that the measures were done at 7 am and 7pm after 7 weeks of diet. In Figure 3a there is no description about the conditions of sampling and the data are very variable from 50nM to 170nM in WT males, i.e. in between the data of am and pm provided in Suppl 3c. Thus, data from Fig 3a may reflect circadian variations from a non-time-controlled blood sampling. The animals with corticosterone concentrations above 150nM in figure 3a look like stress concentrations, the same in Suppl Figure 3c concentrations are expected to be less than 50 nM for am and less than 150 nM for pm in unstressed mice (see eg Gray 2015 PMID: 25559007). The conditions of mice housing conditions, euthanasia and sampling should be described in the Method section: time of light on/off, time of the day for blood sampling in each experiment, time between handling the animal and blood collection (if > to 3 min = stress levels), conditions of euthanasia= use of analgesics/anesthetics or not (this greatly influences cort levels).

We apologise for not providing clearer information on the housing and sampling conditions. We have now included the requested details (Ln 531-534 and 549-552), but would like to elaborate further here for clarity. The reviewer is correct in that

Supplementary Figure 3c presents total corticosterone data in which blood was sampled at the circadian nadir (0700h) and peak (1900h), via tail venesection from conscious, unstressed mice. These diurnal data are in accordance with the literature, with mean AM values of 16.3 nM and 38.3 nM for WT and ELA-KO respectively, and mean PM values of 247.9 nM and 213.6 nM for WT and ELA-KO respectively. The data presented in Figure 3a represents total corticosterone sampled from trunk blood collected following decapitation between 0800h and 1100h. Mice are handled for <1min prior to decapitation, and thus we are confident these are unstressed readouts. In all instances we avoid any use of anaesthesia due to the known impact on HPA activation and thus corticosterone. We have previously published data indicating that mice who undergo restraint stress exhibit circulating corticosterone levels >400nm (and in some cases >1000nM) immediately post-stressor (PMID: 27647861). This is in accordance with other groups who have published similar findings. We believe part of the issue when comparing systemic steroid levels across publications relates to different methodological approaches to quantification. Previously radioimmunoassay and ELISAs have been routinely utilised for determination of circulating glucocorticoids, especially in low volume sampling such as those from tail venesection, however the recognised gold standard for steroid quantification is now via LC-MS/MS, which is the method we include here.

Lines 150-151 fig 3f-3g: p=0.07 so adipocytes tend to be reduced in size in male ELA-KO vs WT. PPARg gene expression may be interesting to measure in the groups.

We agree and have performed this additional measurement, including the new data in Figure 3j and referencing this in the text (Ln 165). Interestingly, *Pparg* expression is elevated in the gWAT of ELA-KO vs WT mice, which may be part of a complex adaptive response aimed at maintaining metabolic balance in gWAT.

Lines 145-147: "in males" should be added in the sentence as females were not tested for these measures

We have added in the suggested text to clarify this was tested in males.

Figure 3: how are CBG Bmax and affinity within adipose tissue?

We thank the reviewer for raising this as it is an important point for advancing research of CBG and glucocorticoids within tissues. However, currently it is not possible to measure CBG Bmax or affinity within tissues, particularly in the absence of blood contamination.

Line 154 and Fig3i: Why are Per1 levels so variable in WT? this is unusual and not seen in liver of the same animal. Fkbp5, Lpl, Atgl & Adipoq are also GC responsive genes but they show no differences between groups, this should be mention in the text. Ccl2 is not significantly different between groups: to be removed in the text as "significant alterations in pro-inflammatory markers"

We have amended the text to clarify that only *Per1* and *Adipoq* were altered as GC-responsive genes. We thank the reviewer for pointing out our typographical error indicating that *Ccl2* was altered, and have amended the text (Ln 164) to indicate that *Cd68* rather than *Ccl2* was reduced in male ELA-KO mice compared to WT controls. As the most reliable GC-responsive gene in adipose tissue (PMID: 27535620), the variability of *Per1* transcript levels in gWAT may simply reflect the variability seen in tissue corticosterone levels. *Adipoq* is significantly increased (in support of reduced corticosterone action).

line 173: "having demonstrated an association": please rephrase as one cannot "demonstrate" an association. Furthermore, the association between NE-deficiency and reduced glucocorticoid action is an overstatement at this stage.

We have rephrased this line as requested.

Line 173-189: adrenalectomy is a very unspecific way to assess the importance of glucocorticoids in the metabolic protection of ELA-KO male mice. Given the question under study (NE/AAT/CBG axis), one would expect an experiment were CBG would be specifically modulated in vivo by a specific knock-down (eg through shRNA) or the use of CBG KO.

Manipulation of systemic and tissue glucocorticoids presents a unique challenge given the multiple feedback mechanisms in place, particularly the HPA axis, that aim to correct any subtle manipulations. We and others have previously utilised adrenalectomy as an approach to determine the importance of glucocorticoids in metabolic and neuropathological conditions (PMID: 27535620, 30292559). However, we are aware of the limitations of this approach and have thus tempered our interpretations to state that the phenotypes observed are adrenal- rather than glucocorticoid-dependent. We have also toned down our discussion and made reference to these limitations (Ln 403-414). As outlined in response to Reviewer 1, we have attempted to carry out further experimentation utilising pharmacological inhibition of NE in HFD-fed mice, however this was hampered by ineffective NE inhibition which made subsequent analysis futile. Manipulation of CBG was another option that we considered. However, given the literature on CBG-KO mice, in which total corticosterone levels are significantly reduced, in parallel with an insufficient glucocorticoid response to stress and tissues presenting with apparent hyporesponsiveness to glucocorticoids (PMID: 20022933, 16980625), we do not believe this approach would help delineate whether NE drives metabolic dysregulation through increased tissue glucocorticoid exposure as it introduces confounding regulation.

Figure 4: the statistics are not provided for graphs 4a, 4c. For the other graphs of Figure 4, interaction and main effects of the 2 way ANOVA are not provided. Interaction has to be significant before running post-hoc tests. Are ELA-KO sham and ELA-KO ADX significantly different for weight gain? AUC Glucose or HOMA-IR? If not the removal of adrenals (and glucocorticoid levels even less) does not play a role for these parameters.

We apologise for not providing more clarity on the statistics used within this Figure, and have now included additional information within the Methods (Ln 718-724) and figure legends to clarify the statistical approaches and the use of post-hoc multiple comparison tests. As requested, we have now included the results of the overall two-way ANOVAs used, providing these in the form of text accompanying the graphs within the revised Figure 4. Moreover, we have included information on the *P*-values in the presentation of these results in the main text (Ln 194-196 and 200-202). These results now show that while for body weight gain there is no difference between ELA-KO sham and ELA-KO ADX, there are significant differences between these two groups for GTT AUC and for HOMA-IR. We thank the reviewer for pointing this out.

Lines 183-184: this assertion is not convincing: it is not significant in sham ELA-KO while it is significant in all other groups (4e); HOMA-IR does not seem different between sham and ADX ELA-KO.

We agree with the reviewer regarding the assertion around the insulin response, and have amended the text accordingly to remove this. Regarding HOMA-IR, as outlined above we have now included our full statistical results, both in the Figure and in the main text (Ln 200-202).

Line 193: "having established.." is an overstatement.

In relation to this line in the results, we understand the reviewer's objection to the term 'established' and have rephrased to state that our murine data demonstrate the link between NE deficiency and glucocorticoid action.

Line 198: Isn't there a word missing in the sentence?

The reviewer is correct, we have added in 'and' within this sentence.

For the clinical study, one would expect that males would be preferentially recruited given the animal data, however 75% of AAT patients were females. Is there a sex difference in the prevalence of the disease or the recruitment was done before the animal data were available? Although BMI and fasting insulin are not different between controls and AAT patients statistically according to Table 1, AAT patients have on average a BMI of obese and show hyperinsulinism. Insulin down-regulates CBG as shown by Fernandez-Real (PMID: 12364459). Is there a correlation between insulin and cortisol levels or BMI and cortisol levels in the subjects, especially the AAT patients? Il-6 is another factor that regulates CBG and might be elevated in AAT patients given their BMI. Its influence on CBG and cortisol needs to be tested.

The reviewer is correct in that recruitment for the clinical study, which was carried out under a 'first come' approach from Generation Scotland, was completed before the outcome of the murine studies was known. To our knowledge, there is no known sex

difference in the prevalence of AAT deficiency. Unfortunately due to the small number of males recruited, it is not possible to undertake any robust statistical analysis of our data to assess potential sexual dimorphism. However, we have now performed regression analyses to determine if Sex predicts free cortisol in our study, and did not find any significant effect ($P = 0.3239$). We have included the results of these analyses in our new Supplementary table 1. Moreover, in our new Supplementary material we have provided graphical representations of all the data generated by the clinical study, including sex-specific presentation of the baseline characteristics of the subjects (Supplementary Figure 7), the arterio-venous tracer study (Supplementary Figure 10), and the CRASH study (Supplementary Figure 13), as well as including discussion within the main text (Ln 391-401). Regarding potential difference in BMI and insulin (Table 1), we thank the reviewer for highlighting this. As discussed in our response to Reviewer 1, we understand that the initial tabular format of the data does not provide sufficient information and could be interpreted as the AAT heterozygotes having a higher BMI and fasting insulin levels. We have now provided graphical representation of the data presented in Table 1 (Supplementary Figure 5). We believe the new graphs, combined with our robust statistical analyses (Supplementary Table 1), now clarify this issue. We thank the reviewer for the suggestion to quantify IL-6, and have indeed measured this in our subjects. Perhaps unsurprisingly, circulating IL-6 levels are very low in all subjects, to the point that 29 out of 32 subjects had IL-6 levels below the limits of detection via ELISA (LLOD 15.6pg/mL). Of the remaining 5 subjects (range 19.3 – 82.9 pg/mL), no statistical relationship was found with either BMI, Sex, fasting insulin, CBG, or free cortisol. While we have not included this data within the revised manuscript, we are happy to add if the reviewer and/or editor feels it is beneficial.

Line 206 onwards: from here, it is very difficult to jump from the text to figure 5 as in the text the placebo arm only is considered but on the figures all groups (placebo and CRASH treated patients) are shown. The CRASH data are presented in the text much later, after figure 6, this is very confusing. Again, as in Figure 4, no stats are provided for graphs 5a, 5b and 5c so one cannot see whether "circulating free fraction is significantly elevated in AAT-deficient heterozygotes" (line 208). Details of interaction and main effects, then post hoc tests should be provided.

We apologise for the confusion in our initial layout and explanation of the results. In our refined manuscript, we have altered the order in which we present the clinical data to make it easier to follow, with the reordered Figures now logically following the main text (Ln 232 onwards). We also apologise for the lack of clarity on our statistical approach and have now provided the statistics for Figure 5 (new Figure 6) as requested, detailing the effects and corresponding P-values on the graphs.

Figure 5e: the data for LPL are not very convincing as 3 AAT patients only (over 16) have high levels. Similar to Figure 3i, glucocorticoid responsive genes such as ATGL or ADIPOQ are glucocorticoid targets but show no differences between groups. One or 2 AAT patients seem to have elevated levels of all transcripts except Adipoq: could this be explained by sex (as there are only few males in the patient cohort) or

by AAT levels (a correlation could be done between AAT levels and levels of the various transcripts).

Our previous studies in human adipose tissue have indicated that short-term (<6h) glucocorticoid treatment robustly induces *PER1* and *LPL* (PMID: 27535620), and the relatively small magnitude of change here may reflect the modest increase in adipose cortisol levels observed in the AAT subjects compared to controls (Figure 6d). As suggested by the reviewer, we have performed additional analyses. Below we highlight the male subjects in our adipose transcript data (Response Figure 4, males highlighted with blue symbols), however there is no indication that sex explains the transcript response in our AAT heterozygotes group. We have also performed regression analyses between the AAT levels and the adipose transcripts, but found no significant correlation.

Response Figure 4. Transcript expression in subcutaneous abdominal adipose from subjects with heterozygous mutations in *SERPINA1* (AAT+/-; orange triangles) and matched controls (grey circles). Male subjects are highlighted in blue symbols. N=4 males in Control group, N=2 males in AAT+/- group. Total N=16 per group. Data were analysed by unpaired t-test. *P<0.05 vs Control.

Figure 6: the figure legend does not fit with the figure and the text, which is very confusing. It might be useful to add "plasma" and skeletal muscle" above the corresponding graphs in addition to correct the legend. Plasma: Two AAT patients have high free cortisol levels: are they males? How are their insulin levels? free cortisol fraction is not shown, is it because the difference between groups is not significant? Skeletal muscle: this time free cortisol fraction is shown (and convincingly different between groups)) but what about free cortisol levels? how are CBG B max levels in this tissue?

We apologise for the incorrect figure legend for the original Figure 6, and for the accompanying confusion from the main text, and thank the reviewer for their suggestions on improving the clarity. We have amended the legend and incorporated their suggested headings into the revised figure (new Figure 5). Moreover, our revised manuscript now

provides additional analyses presenting the data from the clinical studies by sex and by AAT genotype. Of note, the two AAT heterozygous subjects with high free cortisol are females (Supplementary Figure 10). The free cortisol fraction was presented in the original Supplementary Figure 6d, and remains there in the new Supplementary Figure 8d. Unfortunately as outlined in a previous comment we are unable to assess CBG Bmax within tissues due to technical limitations. However, we have also added the free cortisol levels quantified across skeletal muscle as requested, which we present in Supplementary Figure 8e.

Line 304: no correlation analyses were done in the study

In relation to this line in the discussion, we understand the reviewer's objection to the term 'correlates' and have rephrased to state that changes 'are associated' instead (Ln 335)

Line 322: no data in adipose tissue are presented in the manuscript.

We apologise that this was unclear in the text, however in this section of the discussion, we are referring to the data obtained from our placebo/CRASH protocol in the clinical study, in which we obtained adipose biopsies and quantified glucocorticoid levels and transcripts. We have now added some headings to the graphs to clarify for readers (Figure 6d-e).

In the method, serum CBG binding capacity is described twice: (lines 411-426 and lines 601-613) each time mentioning mouse and huma assays. However, in the 2nd description supernatants were incubated with 2 nM of cortisol while it was 20 nM in the first description for human serum. For the statistics, the post hoc tests are not specified.

We thank the reviewer for bring this to our attention. The initial description was aimed at providing details of CBG Bmax determination in mouse and human serum that had been treated with a combination of NE and AAT, while the second method described physiological determination of CBG Bmax directly in mouse and human serum. However, appreciate this was not made clear in the text and have refined this accordingly, combining the CBG binding capacity methodology into a single section (Ln 682-698). Moreover, we thank the reviewer for pointing out our typographical error relating to the concentration of [³H]-corticosterone and [³H]-cortisol, which has been corrected to 2nM in both instances.

Reviewer 3:

In figure 1 the authors should block elastase activity (in NE treatment) in panels a/b and assess the effect on binding capacity.

We have now carried out the suggested experiment and incorporated the results within the revised Figure 1 (Figure 1b and 1c).

Neutrophils are the dominant source of Elane, yet I cannot find details on circulating and tissue counts of neutrophils in any of their models. These should be provided throughout the study.

We agree with the reviewer that assessment of neutrophil numbers would be a helpful addition to our datasets. We have now included neutrophil counts from the human study, as outlined in our revised Table 1. Moreover, in line with one of the reviewer's subsequent comments, we have quantified circulating NE activity in our human samples (Table 1 and Supplementary Figure 5g). Unfortunately, quantification methods for neutrophil counts in murine circulation (cytospin, FACS) requires freshly isolated blood which we do not have for our murine model.

In line with the above comment, neutrophil counts (and the release of Elane) is circadianly regulated. Do tissue corticosterone change over the day and are these in phase with local neutrophil counts?

This is a particularly intriguing point and we thank the reviewer for raising it. In the circulation glucocorticoids levels peak at the start of the 'active phase, usually after waking. In humans, the circadian peak occurs early morning (approx. 0800h) vs in mice where it occurs in late evening (approx. 2000h). Analysis of blood neutrophil counts indicates that in humans, these peak between 2000h and midnight (during the 'rest phase'), while in mice neutrophil counts peak around midday (similarly during the 'rest phase') (PMID: 29877596). Within tissues, circadian variations in glucocorticoid levels generally reflect that seen in the circulation. However, notably in humans, slow turnover of the adipose glucocorticoid pool suggests that fluctuations in circulating glucocorticoids are not reflected in adipose (PMID: 20631029). To our knowledge, there are no studies determining neutrophil oscillations in human or murine adipose tissue, and our search of the Circadian Expression Profiles Database (<http://circadb.hogeneschlab.org>) produced no hits for searches of *Elane* in human or murine adipose. Future studies would benefit from detailed circadian analyses of immune cell numbers and glucocorticoid action in order to determine if a relationship exists between the two.

In figure 3 levels of plasma lipids (triglycerides, cholesterol) should be provided.

We thank the reviewer for this suggestion, and have now quantified serum triglyceride levels from this dataset (Figure 3f) and included this in our results (Ln 159). Limited sample volume allowed us to assess only one parameter, and based on the established links between glucocorticoids and lipid metabolism, we chose to focus on circulating triglycerides.

For Elane concentration rather than activity is given in many panels (e.g. Suppl Fig 2a). Please also provide activity.

Where possible we have now included NE activity (for example in the clinical study, results outlined in revised Table 1 and Supplementary Figure 5g). However, in our murine studies characterising the responses to HFD (Supplementary Figure 2), we chose to focus on NE protein as it provided an established proxy of both neutrophil numbers (which we were unable to measure given our method of collection) and NE activity. Similarly in our genetic mouse model of NE-deficiency, we confirmed deficiency by quantifying NE protein. Unfortunately paucity of samples prevent further analysis in these experimental mice.

It is a striking finding that the authors' key finding does not seem to hold true in female mice. While this is very interesting, it is disappointing that there is no satisfying explanation. Would the Elane/CBG axis be operational in gonadectomized male mice?

We thank the reviewer for raising this point. In our revised manuscript we have now included data demonstrating that in contrast to male mice, female mice fed a HFD do not increase NE locally within visceral adipose tissue (Supplementary Figure 2). We agree that initially it appeared striking that female mice lacking NE were not protected for HFD-induced metabolic dysregulation. However, subsequent interrogation of the literature revealed a common theme in mice on the C57Bl/6J background, with females relatively protected from the effects of HFD (PMID: 22042005, 24942674). Given our new data, it is intriguing to postulate as to the mechanistic driver behind this sexual dimorphism, and a reduced inflammatory stimulus within female visceral adipose has been reported previously (PMID: 20157318). However, it appears that in this model of diet-induced obesity, the NE axis is blunted, and thus action on CBG/glucocorticoids is negligible.

The correct abbreviation for neutrophil elastase gene in mice is Elane, not ELA.

The reviewer is correct and we have amended any text that is incorrect in this regard, specifically in the Methods section (Ln 524).

Figures panels should be referred to in the correct order. The current order in some of the figures is rather confusing.

We apologise to the reviewer for the confusion in our initial manuscript created by the order in which the clinical study results in particular were described, and by the incorrect legend that accompanied the original Figure 6. This point was also picked up by the other reviewers, and we thank them all for highlighting this issue. We have now revised our presentation of the clinical study in both the text (Ln 232 onwards) and with new Figures 5 and 6. Together with the expanded Supplementary material, we believe the results are now significantly clearer.

Authors' responses

Manuscript ID: NCOMMS-23-43150-T

'The NE/AAT/CBG axis regulates adipose tissue glucocorticoid exposure'

Reviewer 1:

'The authors should be commended for the extensive amount of work that they did. In the revised version, they were able to clarify a couple of inaccuracies and as a result, the data are clearer now. Despite that, some of the key findings remain somewhat puzzling, i.e. (i) the fact that the difference is seen in males only and cannot be reproduced in an independent model; (ii) the fact that AATD subjects seem to be neither obese nor suffer a metabolic syndrome. These discrepancies need to be clearly disclosed and the results from the NE inhibitor experiments should be mentioned in the manuscript.'

We thank the reviewer for their kind words regarding the revised manuscript. In line with their suggestions, we have now included a more thorough discussion of the points raised. Specifically, these relate to the sexual dimorphism in the NE/CBG/GC axis (Ln 400-405), and also the lack of association between AAT heterozygous subjects and adverse cardiometabolic outcomes (Ln 376-389). Moreover, we now include in the revised manuscript the additional mouse data outlined in our initial response relating to the NE inhibitor experiment (Supplementary Figure 6), and a discussion of these results in the main text (Ln 340-347).

Reviewer 2:

'Figure 1d and 1e: the authors have left these data showing a percentage of free cort from 50% to 80% which I believe is not correct or understandable. They explain that this artefact come from the fact that the sera were diluted 1:50 prior to treatment with NE/AAT and determination of CBG Bmax and free steroid levels. They claim that this dilution is required because of "the necessary stoichiometry to facilitate NE-mediated cleavage of CBG" and they give as a reference PMID 2370299. In the latter publication, I do not find any mention of stoichiometry necessary for CBG cleavage by NE that would support the serum dilution. Diluting the serum sample will indeed disturb the equilibrium between bound and unbound steroids. Since the free fractions shown are an artefact, probably due to the serum dilution, I think these data cannot be presented as such and the same experiments should be redone on undiluted serum.'

We apologise that we may have been at cross-purposes with Reviewer #2 in our previous response. We agree that physiological % free glucocorticoids would not be as high as this. However, the experiments reported in Figure 1 were not aimed at measuring physiological free glucocorticoids; instead, they were aimed at testing the effect of NE on CBG. For this purpose we are happy to expand further for clarification. As discussed in the manuscript (Ln 447-454) the presence of alpha-1 antitrypsin (AAT) in serum ensures that cleavage of CBG by endogenous NE does not take place. In undiluted serum, the levels of AAT are orders of magnitude greater than NE (PMID: 24241342; 2370299), such that significant

quantities of exogenous NE would be required to overcome this natural inhibition. Given the source of exogenous NE is limited, we utilised diluted serum to enable CBG cleavage and to therefore test our hypothesis that such cleavage would increase the free fraction of glucocorticoids. To test the effect of dilution itself, we quantified the free cortisol fraction in increasing serum dilutions (Response Figure 1), and in accordance with other independent groups (PMID: 24424442) chose an optimal 1:50 dilution to further test our hypothesis. Indeed in almost every published measure of CBG, including binding capacity (Bmax), dilution of the serum/plasma is required in order to perform the assessment. Therefore, we conclude that NE reduces CBG binding capacity and increases the free glucocorticoid fraction, an effect that is inhibited by AAT. However, we recognise that presenting these results as % free has given the erroneous impression that we are inferring physiological free glucocorticoid levels, and so we have now presented this data in Figure 1 as fold change from vehicle control instead.

Response Figure 1. Human serum from a single healthy volunteer was diluted with 0.1M Tris HCL and total and free cortisol determined by ultrafiltration.

'Figure 3a : The corticosterone levels found (mean around 100 nM?) cannot reflect basal morning levels but most probably cort levels of stressed mice. I believe mass spectrometry vs ELISA method cannot explain these very high levels as claimed by the authors. As a proof, these levels are much higher than those measured in Suppl Figure 3d for AM (WT:16.3nM and ELA-KO: 38.3 nM) by Mass Spec by the same authors. Therefore Figure 3a, 3b and 3c I believe are not correct.'

We agree that the morning samples are not truly 'basal' although we do not agree that they represent stressed values either. We present values from samples obtained in several conditions in the manuscript. In Suppl Fig 3C we show truly basal values (WT: 16.3nM and *Elane*^{-/-}: 38.3 nM) obtained by tail venesection on conscious, unstressed mice at 0700h. In Fig 3a we show higher morning values, highlighted by Reviewer #2, for samples obtained using different methodology (samples obtained from trunk blood following decapitation, with <1 min between handling and decapitation) but these remain lower than either the diurnal peak at 1900h (Suppl Fig 3c; WT: 247.9nM and *Elane*^{-/-}: 213.6 nM) or following

restraint stress (Response Fig 2; WT: 618.3 nM and *Elane*^{-/-}: 643.7 nM). To avoid confusion, in the revised manuscript we avoid referring to the Figure 3a results as 'basal' but instead describe them as morning samples.

Response Figure 2. Male WT and *Elane*^{-/-} mice (10 weeks) underwent acute restraint test (in a tube for 10min). Blood was taken by tail venesection immediately prior to stressor (Pre-stress) and 10 minutes later immediately post-stressor (Post-stress). Plasma corticosterone was quantified by LC-MS/MS (n=8 animals per group).

'Figure 3j: Glucocorticoids being anti-inflammatory isn't it strange that the levels of pro-inflammatory markers (*Tnfa*, *Cd68*, *Itgax*) are down-regulated ? could it be discussed in the discussion together with the surprising up-regulation of *Pparg* while adipocyte size tend to be decreased?'

This is an excellent point. Indeed it appears paradoxical that, given the well-known anti-inflammatory properties of glucocorticoids, a reduction in their tissue levels is associated with a concurrent reduction in pro-inflammatory markers. However, much of the research on the anti-inflammatory actions of glucocorticoids has been carried out with saturating levels of exogenously administered glucocorticoids, and has led to an over-simplified view of their effects. Importantly, accumulating evidence suggests that endogenous glucocorticoids can exert potentiating as opposed to suppressive effects on inflammatory markers, and that this is linked to the 'dose' (PMID: 9927693; 17451463; 19647070; 20398732). We have added these points to our revised discussion (Ln 324-332).

'The authors claim that CBG Bmax cannot be measured within tissue. However, there are publications in which CBG Bmax was measured within adipose tissue: PMID 11507677 and 11855738. In this study the evaluation of CBG Bmax within gonadal fat is essential.'

We appreciate the reviewer's concern regarding determination of tissue CBG binding capacity, and thank them for providing references to a group that has attempted this. We are happy to explain our reservations about the methodology used to assess CBG binding capacity, and thus potentially incorrect interpretation of any results. A major issue is that it is almost impossible to accurately measure tissue CBG Bmax without 'contamination' from the circulation. CBG within tissues is proposed to reside within the extracellular/interstitial space. Sampling this space in rodent models is extremely

challenging, and in murine tissues such as adipose, currently unachievable. If the choice was to sample the tissue as a whole, robust measures of tissue-only CBG would require extensive perfusion of the tissues to remove all circulating contamination. In the two publications cited by the reviewer, one carried out this measurement in tissues collected from rats without perfusion (PMID: 11507677). The second study compared rats killed by decapitation with rats perfused under terminal anesthesia (a known confounder for circulating glucocorticoids) and generated cytosolic tissue homogenates to assess CBG Bmax (PMID: 11855738). However, since CBG is not intracellular this approach is not designed to measure the interstitial compartment. Moreover, the authors of the paper themselves caveat their interpretation with a comment that the CBG in adipose 'may result from incorporation of plasma-carried CBG'. So, while we agree with the reviewer that measurement of tissue CBG binding capacity would be hugely valuable, current technological approaches do not support its quantification. We highlight this limitation in our discussion (Ln 454-456).

'The response to the reviewers about the unspecific action of adrenalectomy regarding glucocorticoids involvement is not entirely satisfactory. The authors agree that manipulation of CBG would be a better option than adrenalectomy. They explain that they did not use this option because CBG-KO display very low levels of glucocorticoids. However, they did not discuss my suggestion of using an adipose-specific knock-down of CBG through virus-mediated expression of a shRNA against CBG at adulthood. In this model the HPA axis would not be disturbed as in the full CBG-KO because the manipulation will be localised within adipose tissue and at adulthood with no time for compensatory mechanisms.'

We thank the reviewer and apologise for not adequately addressing the suggestion of an adipose-specific CBG knockdown model. The most significant point here is that the liver is established as the primary source of circulating CBG, and any contribution of adipose to circulating levels is minimal, meaning that the proposed model would not allow appropriate testing of the local tissue-specific NE action on circulating CBG, and subsequently on tissue glucocorticoid exposure. To elaborate on this, while we agree that there are some published reports of CBG being synthesised in adipose tissue, this is not conclusive. Mar Grasa *et al* (PMID: 11855738) and Gulfo *et al* (PMID: 27323695) used qualitative RT-PCR and real-time PCR respectively to show *Serpina6* mRNA in adipose tissue. However, the Cp values of *Serpina6* transcripts are not shown. To date there are no other reports of CBG expression in adipose. Indeed, our own real-time quantitative PCR analyses of CBG in murine adipose tissue reveal that *Serpina6* mRNA is barely detectable, expressed at negligible levels with Cp values >33. We agree with the reviewer that a manipulation of CBG that does not impact systemic glucocorticoids would be the most robust approach. However, we believe the most appropriate way to interrogate this requires a step-advance in technology rather than a simple genetic approach to remove or knockdown CBG. For example, this might take the form of a novel *in vivo* model with 'labelled' CBG that allows differentiation between intact and cleaved forms of CBG. Indeed this forms the basis of proposed developmental research within our group, but is beyond the remit of the current manuscript. In our revised manuscript, we highlight the limitations

of the current technology and suggest future approaches that will be required to further advance this research (Ln 454-456).

'Figure 6d: adipose cortisol is found increased in AAT patients under placebo but not after CRASH (Control CRASH vs AAT CRASH). When separated by sex (Supp Figure 13), the adipose cortisol seems to be up-regulated in male patients only, whether under placebo or after CRASH. In the rebuttal letter, the response Figure 4 shows that the increase in Per1 and LPL are driven by female patients. Together these data do not look congruent.'

The study was not powered to analyse sex effects, and so we refrained from performing any statistical analysis on these.

The clinical study was not powered to analyse sex effects. Thus the regression analysis showing no significant effect of sex should not be reported.

We have removed the regression analysis for 'Sex' from Supplementary Table 1.

The authors claim that their data "underscore the potential therapeutic opportunities to manipulate glucocorticoids and reduce cardiometabolic risk" but in the rebuttal letter they answered to reviewer 1's remark that AAT patients are neither obese nor predisposed to metabolic syndrome, because of likely compensation for the altered glucocorticoid profile, eg due to reduced whole-body 11bHSD1 activity.

Our inference that the NE/AAT/CBG axis provides a target for potential therapeutic manipulation to alter tissue glucocorticoid exposure is based on the evidence in this paper. While this inference is not further substantiated by observations that AAT-deficient patients are obese or predisposed to metabolic syndrome, albeit there is some evidence for increased trunk fat, neither do these observations refute our inference, not least because AAT-deficient patients often have significant health problems including obstructive lung disease and liver failure, significant confounders when determining if there is an underlying change in adiposity and insulin sensitivity. We have expanded on this in the discussion (Ln 376-389).

Reviewer 3:

'The authors have addressed most of my comments; yet, two questions have not been addressed in full:

1. Neutrophils are the dominant source of Elane, yet I cannot find details on circulating and tissue counts of neutrophils in any of their models. These should be

provided throughout the study. Here the authors argue that blood counts are not available; however, I was specifically asking for tissue counts and given that tissues stainings are shown in figure 3, I assume these sections should still be available.

We thank the reviewer for their comments. We apologise for our misunderstanding of their initial request for neutrophil counts, and as requested have now included IHC quantification of neutrophils from the adipose tissue in our mouse model (Supplementary Figure 5). Importantly these support our tissue NE analysis, with reduced neutrophils in gWAT from *Elane*^{-/-} mice vs WT. Moreover, our neutrophil quantification is line with previous reports (PMID: 23562077) in which mice lacking NE displayed reduced neutrophils in the stromovascular fraction of eWAT (quantified by FACS). We regret that in our human samples, paucity of tissue means we are unable to quantify tissue neutrophils in the clinical study.

2. The correct abbreviation for neutrophil elastase gene in mice is Elane, not ELA. This is still not changed throughout, especially in the figures.

We have now amended all the text, including graphs and figure legends, to ensure that all references to neutrophil elastase are correct, and that neutrophil elastase-knockout mice are now correctly referred to as '*Elane*^{-/-}'.

Authors' responses

Manuscript ID: NCOMMS-23-43150-T

'The NE/AAT/CBG axis regulates adipose tissue glucocorticoid exposure'

Reviewer 2:

1. The authors have now well explained the aim of the measures presented in Fig. 1 and they recognize that presenting their results as % free cort was equivocal. It is indeed better to present these results as fold change from vehicle as they did. To be perfectly unambiguous, the authors should add in the legend of FigS1 that free cortisol (%) values presented do not reflect physiological levels because serum samples were diluted. As the authors wrote in the main text, these data on NE/AAT/CBG axis controlling glucocorticoid action are in vitro evidence.

We agree and have included the requested information in the figure legend.

2. Concerning basal/morning corticosterone levels in Fig3a I still disagree with the authors and I still believed that the data reflect stress values even if the values are lower than those obtained by restrain stress, they just reflect a milder stress than restrain. Consequently, the data of Fig.3b and 3c are also mild stress data. In Suppl. Fig.3d, where basal values are clearly obtained, what are the free cort % and free cort concentrations?

We agree that the values in Figure 3a-c do not represent basal, unstressed, circadian nadir levels. We have revised the manuscript to clarify that the circulating corticosterone levels presented in Figure 3a 'were obtained under cull stress conditions' (Ln 142).

Regarding Suppl. Figure 3d in which we present diurnal nadir corticosterone levels; these blood samples were obtained by tail venesection in conscious, unrestrained mice. In order to collect a sample within <1 min from disturbing the cage, we cannot obtain >40-50µL of blood (and therefore approx. 20-25µL of plasma). To accurately perform equilibrium dialysis and subsequently quantify free corticosterone by LC-MS/MS, we require a minimum volume of 100µL of plasma. Therefore, technical limitations prevent us from assessing free steroid levels in unstressed tail bleed samples. We have included discussion on this in the manuscript and highlight this as a limitation (Ln 349-352).

3. The tentative explanation about reduced proinflammatory markers in Fig.3j is far-fetched. The data remains puzzling. As already discussed in previous reviewing the gene expression data, that are purposed to show increased or decreased glucocorticoid action, are not convincing. The usual glucocorticoid responsive genes examined in many studies are Per1, Fkbp5, and Sgk-1. Here,

Per1 only is tested but highly variable results are presented in WT animals for gWAT tissue so it is not enough. Then pro-inflammatory genes and fat synthesis gene (Pparg) are opposite to expectations. Fkbp5 was measured in sWAT, why not in gWAT?

We understand Reviewer #2's comment regarding a lack of change in glucocorticoid-responsive genes to support our interpretation in gWAT, and thank them for the suggestion to analyse additional targets. We now include quantification of *Fkbp5* and *Sgk1* and show that, in line with *Per1*, expression of these glucocorticoid-responsive transcripts is also reduced in gWAT (see Fig 3j of revised manuscript). Regarding the pro-inflammatory gene expression data, we apologise that we may have been at cross-purposes with the Reviewer in our previous response. We were not stating that the changes in pro-inflammatory genes represent direct glucocorticoid-induced changes. Indeed they are more likely a direct consequence of reduced neutrophil elastase. However, as outlined in our previous response, multiple studies have demonstrated that a decrease in glucocorticoid levels does not always correlate with reduced pro-inflammatory gene expression (PMID: 9927693; 17451463; 19647070; 20398732). Regarding *Pparg*, while we agree this appears paradoxical given reduced adipocyte size, this is consistent with the change in adiponectin and pro-inflammatory transcripts, and presumably reflects complex regulation of *Pparg* beyond an exclusive glucocorticoid effect. Our revised manuscript now includes further discussion of these important points (Ln 325-327 & Ln 331-343).

4. As for CBG Bmax measured within fat tissue, I think mice can be perfused after short isoflurane anesthesia (that do not disturb circulating cort levels) with 1 ml of PBS in the heart to wash out blood contamination to all organs. It would have been worth trying.

We appreciate Reviewer #2's intention to permit measurement of CBG binding capacity from within tissues. However, we do not agree that a suitable method exists, or that the proposed approach using a small volume infusion after a short anaesthesia is appropriate. To perform a whole-body perfusion, mice need to be completely anaesthetised to allow opening of the chest cavity to expose the heart and permit needle insertion into the left ventricle, followed by subsequent perfusion with PBS. A minimum of 10 mL would then be required to sufficiently perfuse the whole body (PMID: 33796622). Using isoflurane, to ensure the mouse reached a surgical plane of anaesthesia by loss of response to tail or toe pinches - a requirement of UK Home Office Animals (Scientific Procedures) Act 1986 - would require >5min in addition to the time for surgery to be conducted. Reports in both mice and rats, as well as our own data, indicate that >3min of isoflurane exposure is sufficient to increase circulating and tissue corticosterone levels (PMID: 34432644, 26946276, 23022994), rendering the interpretation of tissue CBG binding questionable. We agree that this is an important issue and in our revised manuscript, we now discuss these points and highlight that measurement of CBG binding capacity in tissue remains a limitation of this work (Ln 462-471).

5. Concerning CBG inhibition within fat tissue, I now understand that the hypothesis of the authors is that CBG in fat tissues derives from plasma rather than being synthesized in situ. First, this remains to be proven even if the authors do not believe previous published data. Then, I understand that the general hypothesis (in mice) is that although serum-derived AAT is present within adipose tissue the increased NE levels, caused by neutrophil infiltration, overcome AAT inhibition. This is not demonstrated here. Could AAT be measured within adipose tissue? If NE/AAT balance is really increased in adipose tissue compared to serum, could AAT be added specifically within adipose tissue of HFD-fed male mice to rescue the phenotype? Another possibility would be to add the NE inhibitor (GW311616A) locally in adipose tissue of HFD-fed male mice in order to prove the local action of NE in gWAT and increased glucocorticoid exposure.

Yes, this is our hypothesis based on previous literature demonstrating infiltration of neutrophils, the source of NE, specifically into obese adipose tissue (reviewed in PMID: 35741012). Reviewer #2 helpfully suggested measuring adipose AAT levels in response to HFD would provide more robust evidence, and we now include this additional data in our revised manuscript. Importantly, our quantification of AAT in mice reveals that HFD does not alter either serum or adipose tissue AAT levels. Therefore, the HFD only increases the NE:AAT ratio in adipose tissue. We have now included this new data in Supplementary Figure 2h-k and in the main text (Ln 116-119 and Ln 331-333).

However, an additional study to pharmacologically add AAT or GW311616A into adipose would neither be feasible nor be likely to improve the manuscript. To achieve local, targeted pharmacological addition of AAT or inhibition of NE specifically in visceral adipose would require administration via IP injection potentially several times a week for up to 8 weeks (duration of HFD). Such repeated invasive administration carries with it a substantial risk of inducing local inflammation and stress, which would significantly confound interpretation. Moreover, AAT is not commercially available and in our previous response to reviewers we presented our use of the reported NE inhibitor GW311616A which failed to inhibit NE (in 2 separate studies, one of which is included in Supplementary Figure 6).

6. Such NE increase in gWAT of male mice fed HFD is not observed in females fed the same diet. However, I wonder if this does not reflect a floor effect since females show much reduced (if any?) metabolic dysregulation after 8 weeks of HFD consumption. Since there is no comparison with females fed a standard chow, it is difficult to conclude. A longer exposure to HFD might reveal similar NE increased in females too.

Our study aimed to compare males and females at equivalent time points/ dietary durations to determine if NE deficiency was protective in both sexes. As Reviewer #2 highlights, neither was NE deficiency protective in female mice nor did HFD for 8

weeks lead to increased NE in gWAT in female mice. We agree that this result is difficult to interpret given that female mice are less prone to HFD-induced metabolic dysregulation (insulin resistance), a sex difference observed in multiple other studies (PMID: 20157318, 22221155, 24942674, 22662224). As discussed in our manuscript, there are likely factors beyond NE/glucocorticoids that play a role in the sexually dimorphic response to HFD. Following the same reasoning as the Reviewer, we have completed a study administering 16 weeks of HFD or chow diet in female mice, comparing GTT response at 8 weeks, equivalent to our manuscript study (Response Figure 1a-b), and quantifying NE in gWAT at 16 weeks (Response Figure 1c). We find that female mice do develop mild metabolic dysregulation, evidenced by impaired glucose tolerance, but even at 16 weeks of HFD exposure do not exhibit an increase in adipose NE compared to chow-fed controls. We do not see this as a key piece of data to add to the manuscript. However, in our revised manuscript, we include further discussion of the sex-specific responses to HFD in mice (Ln 420-424) and include the discrepancy between the mouse and human data with regards to sex (Ln 433-435).

7. It is still puzzling that in AAT+/- patients, free cort % is increased in the circulation but not corrected by the HPA axis feedback mechanism. Furthermore, CBG levels are unchanged but they predict free cort levels by regression analysis. How can this be explained?

We agree that the apparent inability of the HPA axis to 'correct' the increase in free cortisol is intriguing. Indeed, an important caveat is that our measurements are made in venous blood from the hand, albeit arterialised by hand-warming, which may not be representative of capillary blood and interstitial fluid at HPA axis feedback sites (hypothalamus and pituitary). In our revised manuscript, we have expanded our discussion to include this explanation (Ln 374-377).

Regarding the relationship between CBG and free cortisol, we believe the Reviewer is referring to CBG binding capacity (Bmax) correlating with free cortisol (Supplementary Table 1). Using an unpaired t-test, CBG Bmax values between the two groups (AAT+/- subjects and controls) are not statistically different, while free cortisol levels are significantly correlated with CBG Bmax by regression analysis. The regression analysis was conducted at Reviewer #2's request on the whole data set without adjustment for AAT genotype, whereas the t-tests were used as part of our original analysis plan to compare between AAT genotypes. The regression analysis is consistent with a relationship between AAT and free cortisol, but this was not corroborated by the different approach of t-tests. In our revised manuscript we now highlight the regression analyses regarding free cortisol and CBG Bmax (Ln 474-477).

8. In Fig.6d the increased adipose cortisol is driven by 4 patients out of 16. For Per1 and LPL transcripts the increased seems also driven by few patients. Are they the same patients for cort and transcripts high values? Could a correlation be calculated? If significant this would mean that another factor than AAT influence cortisol levels in AAT+/- patients.

We respectfully disagree with Reviewer 2's statistical interpretation. It is common for two different groups to have overlapping distributions but for there to be a statistical difference between them (i.e. significantly different means in their Gaussian distribution). We do not think that further correlation analysis is appropriate but, to allay Reviewer #2's concerns, we have conducted an exploratory analysis, calculating Pearson's Correlation Coefficients. These demonstrate that the patients 'with high cortisol' are not the same patients who have 'high transcript values' (Response Figure 2a-c). However, we agree that measurement of additional transcripts would provide more robust evidence in line with our murine study (see point 3 above) and we now demonstrate that *SGK1* is also increased in AAT+/- subjects compared to controls (Fig 6e), further strengthening our conclusions that glucocorticoid-responsive genes expression is altered in the adipose tissue of AAT heterozygous patients.

Response Figure 2. a-c, Pearson's Correlation analysis of adipose cortisol vs adipose transcripts *PER1*, *LPL*, *SGK1* (significantly upregulated glucocorticoid-responsive genes) from subjects with heterozygous mutations in *SERPINA1* and matched controls.

9. The abstract does not reflect the data obtained in the study. The abstract does not reflect the data obtained in the study:

We apologise for any lack of clarity in our original abstract. Following the Reviewer's advice and suggestions, we have thoroughly reworked our abstract to better reflect the data.

- 1) ***it is odd to read the sentence "Here we used complementary approaches in mice and humans to show that male mice lacking NE fed a high-fat diet have reduced glucocorticoid exposure in adipose tissue, with improved glucose tolerance and insulin sensitivity". The most important findings are not presented and the females results ignored.***

We have now highlighted that sexually dimorphic nature of our murine study, and focused the abstract on the HFD-driven increase in NE (and the NE:AAT ratio) specifically in visceral adipose.

- 2) ***“The protective effect of NE deficiency is lost when endogenous glucocorticoids are removed” is not correct since whole adrenals are removed. Furthermore, this result is not a major point in the story.***

We have corrected this to state that that the effect is lost when adrenals a removed.

- 3) ***Yes AAT+/- patients seem to have higher circulating free cort but the latter is unexplained given than neither CBG nor feedback control are impacted.***

In our main text, we have now included further interpretation and discussion of the relationship between the results in the human study. In our new abstract, we have highlighted the effect of AAT deficiency on adipose glucocorticoid levels and transcriptional action, and included the contrast with the mouse data in which AAT deficiency results in changes in circulating free cortisol independent of HPA axis activation.

- 4) ***Yes adipose and skeletal glucocorticoid exposure look increased but maybe only in part of patients (4/16). The mechanism of this increased exposure is different from mice as the free cort fraction is increased systemically.***

We address this in point 8 above, but have also included in the abstract a statement outlining the contrast between the mouse and human studies in terms of tissue glucocorticoid exposure.

- 5) ***“These data demonstrate that NE and AAT modulate local tissue glucocorticoid bioavailability in vivo, revealing a novel mechanism linking inflammation and metabolism. » To me, as discussed above, there is no data to claim such “demonstration” of “local tissue glucocorticoid bioavailability regulation by NE and AAT”.***

We have reworded this sentence in line with our new data.

In my opinion, what can be claimed at this stage is that adipose cort levels are increased in gWAT of HFD-fed male mice compared to standard chow fed animals and associated with increased NE specifically in this tissue. Additionally, NE total ablation (Elane-/- mice) results in reduced cort levels in gWAT specifically. Together these 2 results are indeed very interesting. It remains to demonstrate that 1) NE/AAT levels in gWAT are sufficiently increased so that AAT no longer inhibit NE action on CBG and that cleavage of CBG within adipose (and maybe muscle) is increased too. In humans the increased cort release in adipose and muscle of (some) AAT-/+ patients are

very interesting but as discussed above further data are necessary to be convincing.

We thank the Reviewer for providing a clear summary of their points. We believe that, as outlined in our responses above, the new additional data and interpretation ensures that our revised manuscript is now suitably placed to provide to make a valuable and timely contribution to the field of endocrine metabolic research.

Authors' responses

Manuscript ID: NCOMMS-23-43150-T

'The NE/AAT/CBG axis regulates adipose tissue glucocorticoid exposure'

Reviewer #2 (Remarks to the Author):

The additional data on glucocorticoid-responsive genes expression in both mice and humans is strengthening the hypothesis put forward. The measure of AAT in mice is also new data in favor of the hypothesis. While I still believe that measurement of CBG binding activity in tissue without blood contamination is feasible, this important limitation is now presented in the discussion. Similarly, AAT supplementation as done in the article D'Souza, R. F. et al. Am J Physiol Endocrinol Metab 321, E560-E570, seems feasible to me and would have provided further functional evidence of AAT/NE involvement. Prolastin-C (human α 1-antitrypsin A was provided free of charge by Grifols USA, Los Angeles, CA in D'Souza 's article.

We thank the reviewer for their comments on the extra data we have provided, and for highlighting that while we have delineated a number of novel associations, there are also a number of limitations to our work. We have taken care to ensure that these limitations are highlighted throughout the manuscript, but particularly in the discussion to ensure appropriate interpretation.

The section on adrenalectomy still needs improvement in the writing because it still read as if adrenal removal was a proof of glucocorticoid involvement:

Eg line 185-186: "we tested whether the metabolic protection of Elane-/- male mice was dependent on glucocorticoids » To me, the authors tested if it was dependent on adrenal presence.

Line 192-193 « Assessment of glucose tolerance revealed a similar outcome, with Elane-/- mice lacking endogenous glucocorticoids having no further improvement in their glucose tolerance compared to WT littermates lacking glucocorticoids (Figure 4c and 4d) ».

Line 206 « Our murine data demonstrated that NE deficiency attenuates adipose glucocorticoid action » : to me there is no demonstration, I would suggest to write : Our murine data showed that NE deficiency is associated with reduced adipose glucocorticoid action »

We have revised the suggested sentences to ensure that 'adrenal steroids' are the focus of the experimental interpretation rather than solely glucocorticoids.

In the discussion, from line 322 it should be mentioned more clearly that the data were found in male mice because as it is, it reads as if it was a general finding.

We agree and in our revised manuscript have included reference to 'male' mice at several points within this discussion section.

There are a number of typo errors, eg line 144, line 1440

The manuscript has been checked for typographical and grammatical errors.